# Ant colonies maintain social homeostasis in the face of decreased density

**Andreas P Modlmeier[1]\*, Ewan Colman[2], Ephraim M Hanks[3], Ryan Bringenberg[1], Shweta Bansal[2], David P Hughes[1,4]\***

[1]Department of Entomology, College of Agricultural Sciences, Penn State University, State College, United States; [2]Department of Biology, Georgetown University, Washington, DC, United States; [3]Department of Statistics, Eberly College of Science, Penn State University, State College, United States; [4]Department of Biology, Eberly College of Science, Penn State University, State College, United States

**Abstract** Interactions lie at the heart of social organization, particularly in ant societies. Interaction rates are presumed to increase with density, but there is little empirical evidence for this. We manipulated density within carpenter ant colonies of the species *Camponotus pennsylvanicus* by quadrupling nest space and by manually tracking 6.9 million ant locations and over 3200 interactions to study the relationship between density, spatial organization and interaction rates. Colonies divided into distinct spatial regions on the basis of their underlying spatial organization and changed their movement patterns accordingly. Despite a reduction in both overall and local density, we did not find the expected concomitant reduction in interaction rates across all colonies. Instead, we found divergent effects across colonies. Our results highlight the remarkable organizational resilience of ant colonies to changes in density, which allows them to sustain two key basic colony life functions, that is food and information exchange, during environmental change.
DOI: https://doi.org/10.7554/eLife.38473.001

\*For correspondence:
andreas.modlmeier@gmail.com
(APM);
dph14@psu.edu (DPH)

**Competing interests:** The authors declare that no competing interests exist.

## Introduction

Ants are one of the most ecologically successful groups in nature: they are widespread and abundant in almost every terrestrial ecosystem, and have been around for >140 million years (*Moreau and Bell, 2013*). This success has been attributed to their social organization, particularly their division of labor (*Wilson, 1971*). The ants' high degree of social organization has allowed them to develop collective behaviors that have many similarities to human societies: ants have complex architecture, have true agriculture cultivating fungi for food and herding aphids, use antibiotics and wage war with each other (*Hölldobler and Wilson, 2009*). Many collective behaviors in ant colonies are presumed to be the result of self-organization, in which complex colony-level patterns emerge from local interactions among workers following simple rules (*Beshers and Fewell, 2001*; *Bonabeau et al., 1997*). The rates of these interactions are particularly important, as they have been shown to influence decision making, task allocation and task intensity in ants and other social insects (*Gordon and Mehdiabadi, 1999*; *Greene and Gordon, 2007*; *O'Donnell and Bulova, 2007*). However, such elaborate organization also calls for constant regulation to maintain the *status quo* or, if necessary, to facilitate an adaptive shift to a new state, so that the colony can sustain essential social processes such as food distribution, thermoregulation, defense and nest construction (cf. social homeostasis; *Hölldobler and Wilson, 1990*).

Interaction rates are typically thought to be density-dependent, implying that changes in colony size and density could significantly alter group dynamics (*Pacala et al., 1996*). Changes in colony

size and density are a naturally occurring phenomenon that are part of the life cycle of wood dwelling social insects such as carpenter ants. Colonies start with a single queen that moves into a small wooden cavity and lays eggs. Once the workers hatch, they have to excavate wood to enlarge the nest and allow for colony growth. Accordingly, each subsequent brood cycle will lead to fluctuations in colony size and density. Density changes can also occur when workers discover an attractive nesting location and the colony decides to split into multiple nest sites and thus becomes polydomous (*Buczkowski, 2011*). To minimize potentially adverse effects resulting from changes in density and maintain social homeostasis, ant colonies should therefore try to actively manage the rates of their interactions. Indeed, a study that manipulated colony size and density by putting workers in various arena sizes found that ants may be able to regulate their interaction rates (as measured by counting contacts via antennae) by forming clusters when density is low or by avoiding contact with nearby neighbors if density is high (*Gordon et al., 1993*). However, any change in the distribution of workers in the nest could substantially alter a colony's spatial ordering of work.

Workers are known to exhibit distinct movement zones, that is 'spatial fidelity zones' (*Sendova-Franks and Franks, 1995*), which have been linked to an individual's behavioral repertoire in a variety of social insects (*Baracchi and Cini, 2014*; *Heyman et al., 2017*; *Jandt and Dornhaus, 2009*; *Mersch et al., 2013*; *Powell and Tschinkel, 1999*; *Robson et al., 2000*; *Seeley, 1982*). This is thought to play a key role in the development of the division of labor by promoting task specialization and reducing task switching costs (*Bourke and Franks, 1995*). It is therefore important not only to study how changes in density influence the interaction rates of a colony, but moreover to consider how ants alter their spatial and social dynamics when density changes. Surprisingly, there has been a lack of studies that have combined spatial and social network statistics to examine how social insects maintain homeostasis after intracolonial density changes. In our experiment, we concentrated on one of the most important social interactions in ants and other eusocial insects: trophallaxis. Its primary function, the fast and efficient transfer of liquid food via regurgitation among colony members, is crucial because only a few workers, the foragers, leave the nest to collect resources, so trophallaxis ensures that all colony members receive food.

## Results and discussion

We manipulated the nests of the common black carpenter ant *Camponotus pennsylvanicus* to examine the effect of intracolonial density on spatial organization, food transfer, and social connectivity. To do this, we filmed three colonies under high-density conditions for four hours each (*Figure 1A*, *Figure 1—figure supplement 1*) before quadrupling the available nest space. Following a week of habituation to this low-density setting, we recorded four additional hours under low-density conditions, leading to a total of 24 hr of video footage. The videos were watched multiple times resulting in about 630 hr of observations of trophallaxis and 980 hr of observations of tracking. From these videos, we manually recorded 6,912,480 ant locations and 3262 trophallaxis events. We recapitulated the natural conditions of the wood-nesting carpenter ant, *C. pennsylvanicus*, by filming ants inside completely dark wooden nests that were connected to foraging arenas, providing access to food and water located 188 cm away from the colony. Each colony was comprised of a queen and between 77 to 85 uniquely labeled workers and 15 larvae.

We first investigated how the four-fold increase in nest space influenced the ants' overall distribution and spatial organization inside the nest. Instead of evenly dispersing across the newly gained space, which now consisted of four connected chambers instead of just one chamber, the colonies split into two spatially separated groups. During the observation period, about 39% of all ants in the nest (38.8 ± 2%, 37.2 ± 2%, 42.4 ± 2%; average percentage ± SD of colonies 1–3, respectively) aggregated in the entrance chamber, while around 60% (60.4 ± 3%, 62.6 ± 2%, 56.2 ± 2%), which included the queen for each colony, stayed in the chamber farthest away from the entrance, henceforth referred to as the queen chamber. Only about 23% (30.6%, 20.3%, 18.3%; colonies 1–3, respectively) of all workers moved between the two ant groups in the entrance and queen chamber (*Figure 1—figure supplement 1*, *Figure 1—figure supplement 2*).

We next analyzed whether the increase in nest space influenced spatial organization, that is how ants organize themselves into distinct groups on the basis of their space use in the nest. To divide the colony into distinct groups, we first quantified the position in the nest of each ant at every point in time (*Figure 1B*). We then measured the similarity between each pair of ants on the basis of their

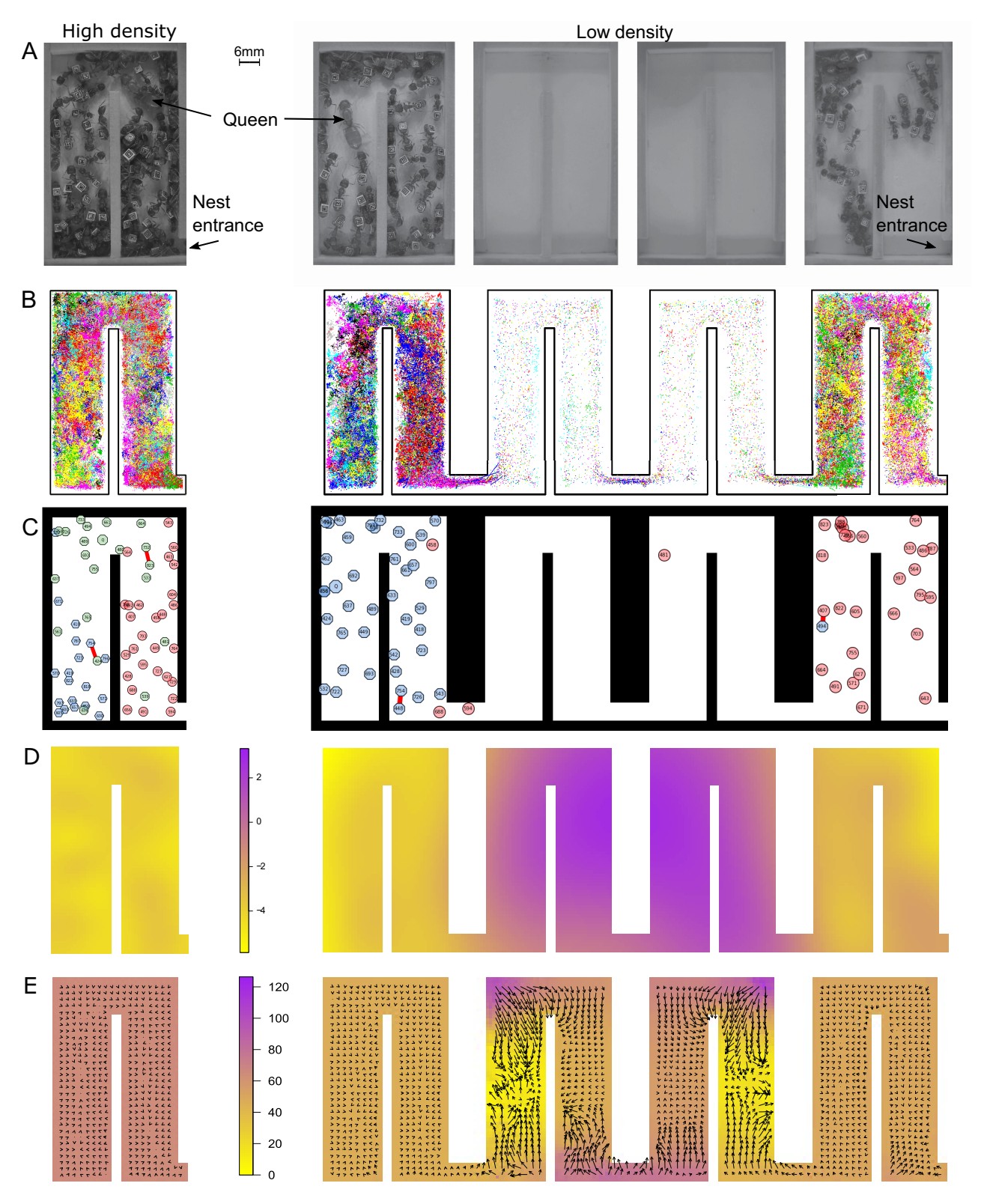

**Figure 1.** At low density, colonies exhibit dramatically different movement patterns in the center part of the nest that result in faster travel between groups. (**A**) Still images from the videos that were used to track ants in colony 1 in high- (one chamber) and low-density (four chambers) environments. (**B**) Tracking data for all ants in colony 1. (**C**) Still images from the animations visualizing ant identity (each labeled polygon represents a specific ant),

*Figure 1 continued on next page*

spatial signature. Finally, we applied a community detection algorithm to partition this network into groups of similar ants (see 'Materials and methods'). Our algorithms revealed two to three distinct spatial groups inside each nest (*Figure 1C*). Ants that were outside the nest during the entire observation period were added as an additional spatial group. We found that an ant's spatial group before the nest expansion predicted its spatial group afterwards (*Figure 2A* and *Figure 2—figure supplement 1*; Spearman's rank correlation: r = 0.26, p = 0.03; r = 0.47, p<0.0001 and r = 0.23, p = 0.03; colonies 1–3, respectively). Thus, the fourfold increase in space merely allowed the groups that were already present within each colony to separate from each other.

Spatial organization was also related to task performance, suggesting that the colonies were able to maintain their spatial ordering of work. On average, 95% of all foragers were part of the spatial group located closest to the nest entrance or outside the nest. By contrast, the one or two groups in the back of the nest harbored the queen and brood, presumably allowing the colony to form a line of defense against potential attacks by enemies. Overall, our results emphasize the remarkable resilience of ant colonies against disturbance, allowing the ants to conserve their relative spatial grouping in the nest despite the shift in the actual spatial organization (i.e., the new use of a high-speed, low-occupancy 'transit' section).

To study the relationship between spatial organization and interaction patterns, we calculated the assortativity, that is compartmentalization, of the trophallaxis network with respect to the spatial grouping that our classification algorithms revealed (*Colman and Bansal, 2018*; see 'Materials and methods'). Assortativity measures the tendency of individuals to interact with others from the same group rather than those from other groups (*Newman, 2003*). Owing to the spatial fidelity of individual ants, we expected positive assortativity, because ants can only interact with ants that are close to them. We indeed found that assortativity was significantly positive across all colonies and treatments (more than two standard deviations above 0 using the jackknife method described in *Newman, 2003*). This suggests that each colony consists of a network of tightly knit spatial groups with dense connections within a group but sparse connections between groups (*Figure 2B*; *Figure 2—figure supplements 2*, *3*, *4*, *5* and *6*; *Video 1*). In such a compartmentalized network, the regulation of colony activity presumably relies on a few key individuals that form connections between spatial groups (*Fewell, 2003*; *Richardson et al., 2018*). We reason that ants may mitigate the effects of spatial separation on their social network by preserving their spatial organization and by reducing travel time between spatial groups.

Although assortativity can modify the connectivity within a colony, it does not take overall interaction rates into account. Previous findings had shown that encounter rates increase linearly with density, but level off when densities are high (*Gordon et al., 1993*). We found that trophallaxis interaction rates did not increase with global density (*Figure 2C*). This is in agreement with the results of *Gordon et al. (1993)*, who measured contact rates via antennation in an open arena. The significant interaction between colony and treatment suggests divergent trends among colonies (*Table 1*). Indeed, a further pairwise comparison of the interaction terms (*Table 1—source code 1*; *Table 1—source data 1*) revealed that for colony 1 and colony 2, there is significantly less trophallaxis during high-density than during low-density periods (*Table 1—source data 2*). For colony 3, there is no significant difference in the trophallaxis rates between low- and high-density settings.

This experiment shows that — contrary to expectations — ant interaction rates did not decline after we decreased density. There are several plausible explanations for this finding: a change in density may have little effect on interaction rates, because they are limited by physical and physiological constraints (*O'Donnell and Bulova, 2007*). This is particularly relevant for interactions like trophallaxis, in which an

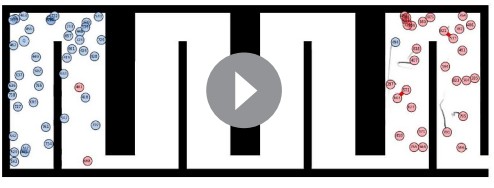

**Video 1.** Animation for colony 1 depicting the combined tracking and trophallaxis during the low-density period for about 1 hr of data. Ants are represented as circles with their identity (numbers identify workers, 'Q' represents the queen). Food sharing is visualized as a red line between two individuals. Circle color indicates spatial-group affiliation. Video runs at 10 times normal speed.
DOI: https://doi.org/10.7554/eLife.38473.014

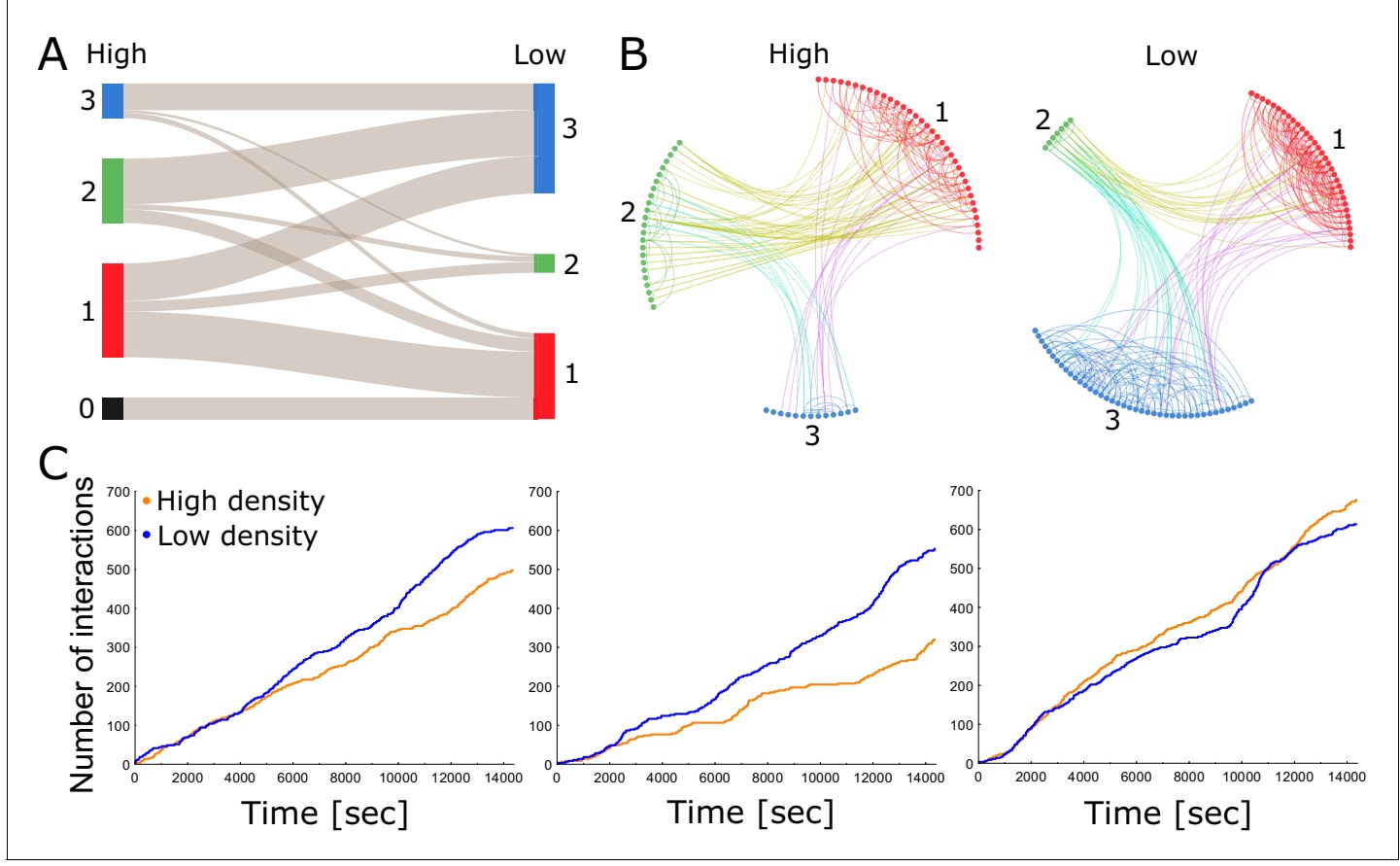

**Figure 2.** Each colony consists of a network of tightly knit spatial groups that exhibits remarkable resilience to changes in density. (**A**) The colored columns depict the different spatial groups identified by the clustering algorithm in the high- and low-density treatments of colony 2. Low numbers represent a closer position to the entrance. The gray lines between these groups visualize that the ants preferred to remain in a similar spatial position relative to each other when nest size was increased. (**B**) Trophallaxis networks of colony 2 before (high density) and after nest expansion (low density). Each node represents an ant in the network with color indicating its spatial group. Groups are arranged in a circle and sorted numerically on the basis of their average distance to the entrance. Specifically, group 1 was closest to the entrance and group 3 furthest away. (**C**) Cumulative number of interactions (food sharing events) over time during high- and low-density periods for colonies 1, 2 and 3 (from left to right).
DOI: https://doi.org/10.7554/eLife.38473.007

The following figure supplements are available for figure 2:

**Figure supplement 1.** The colored columns depict the different spatial groups identified by the clustering algorithm during the high- and low-density periods for colonies 1, 2 and 3 (from left to right).
DOI: https://doi.org/10.7554/eLife.38473.008

**Figure supplement 2.** Trophallaxis networks of all colonies before (high density) and after nest expansion (low density).
DOI: https://doi.org/10.7554/eLife.38473.009

**Figure supplement 3.** Location of trophallaxis (food sharing) events during the low- and high-density periods.
DOI: https://doi.org/10.7554/eLife.38473.010

**Figure supplement 4.** All locations of ants in colony 1 by spatial group and density treatment.
DOI: https://doi.org/10.7554/eLife.38473.011

**Figure supplement 5.** All locations of ants in colony 2 by spatial group and density treatment.
DOI: https://doi.org/10.7554/eLife.38473.012

**Figure supplement 6.** All locations of ants in colony 3 by spatial group and density treatment.
DOI: https://doi.org/10.7554/eLife.38473.013

individual can only share food with one or, in rare instances, two individuals at the same time (personal observation, Modlmeier). Furthermore, ants that frequently share food with each other may have been slower or unable to find their usual interaction partner in the dense crowd. This implies that the form and function of an interaction rate may affect how its rate is affected by changes in the

density of the society. Alternatively, or in addition to this, ants may be able to regulate their interaction rates actively to keep them at an optimal level for the colony by: (a) changing their movement and distribution in the nest (*Adler and Gordon, 1992*; *Gordon et al., 1993*; *Davidson and Gordon, 2017*), thus maintaining local density, or (b) making a change in their interaction behavior that is independent of local (realized) density. To examine the relative support for these two scenarios in our experimental data, we measured the local (realized) density around each ant, here defined as the mean number of other ants within 15 mm (please see 'Materials and methods' for details). We estimated local density to be 9.9 ants in high-density nests and 7.9 ants in low-density nests. This difference was significant (Wald Chi-square test, $p < 0.00001$), indicating that local density was indeed lower for ants in low-density nests than for ants in high-density nests. Also, it seems to be worth noting that even though there is a reduction in local density, this reduction is substantially lower than would be expected under a null model of random diffusion (i.e., a 75% reduction in density), so it still suggests that the ants are regulating local density, to at least some degree.

To get a clearer picture of the effect of local density, we also performed an analysis of when trophallaxis events occurred between pairs of ants. By viewing trophallaxis events as arising from an inhomogeneous Poisson process, we examined how rates of trophallaxis initiations differed across colonies (1,2,3), treatments (high and low density), and also different levels of 'local density'. We defined local density as the number of additional ants (beyond the focal pair) within a range of spatial lags (5 mm, 10 mm, 15 mm, and 20 mm). This allows us to examine how much of the variation in trophallaxis rates can be attributed to differences in colony, treatment, and local density. We found no significant interactions ($p$-value $> 0.05$, Z-test) in this analysis between colony and treatment (high or low density). We found significant interactions between colony and local density (numbers of ants within different spatial lags), but the qualitative patterns in the effects of local density were conserved across all colonies (see *Figure 3*). Furthermore, we found lower rates of trophallaxis in the high-density treatment than in the low-density treatment (*Table 2*). This significant main effect of treatment provides direct support for the idea that the relationship between local density and interaction rate differs between density treatments, suggesting a change in behavior that is independent of local density. In addition to this treatment-level effect, we found that local density had significant effects on trophallaxis initiation rates, and that this effect varied slightly between colonies. There were significant interaction effects between local density and colony, but the qualitative effect of local density is very consistent across the three colonies (see *Figure 3*): the more ants that are very close (5 mm or 10 mm) to an ant pair, the higher the rate of trophallaxis initiations between pairs,

**Table 1.** Summary of the parameter estimates of a generalized linear model with Poisson distribution and log-link predicting the number of interactions that were initiated each second depending on treatment (high versus low density) and colony identity (1, 2 and 3).

Total sample size is 86,406. 14,401 observation points (one observation per second) per treatment and colony (14,401 × 2×3=86,406). CL, confidence limit.

|  | Level of effect | Estimate | Standard error | Wald stat. | Lower CL 95% | Upper CL 95% | p-value |
|---|---|---|---|---|---|---|---|
| Intercept |  | −3.30 | 0.02 | 33538.95 | −3.34 | −3.27 | <0.0001 |
| Colony | 1 | 0.04 | 0.03 | 1.97 | −0.01 | 0.08 | 0.16 |
| Colony | 2 | −0.23 | 0.03 | 71.58 | −0.28 | −0.18 | <0.0001 |
| Treatment | High | −0.11 | 0.02 | 36.34 | −0.14 | −0.07 | <0.0001 |
| Colony*Treatment | 1 | 0.01 | 0.03 | 0.12 | −0.04 | 0.06 | 0.73 |
| Colony*Treatment | 2 | −0.16 | 0.03 | 36.86 | −0.22 | −0.11 | <0.0001 |

DOI: https://doi.org/10.7554/eLife.38473.015

The following source data is available for Table 1:
Source code 1. R code for the pairwise comparison of the interaction terms colony by treatment presented in *Source data 2*.
DOI: https://doi.org/10.7554/eLife.38473.016

Source data 1. The Source data for the results presented in *Table 1* and *Source data 2*.
DOI: https://doi.org/10.7554/eLife.38473.017

Source data 2. Pairwise comparison of the colony by treatment interaction terms presented in *Table 1*.
DOI: https://doi.org/10.7554/eLife.38473.018

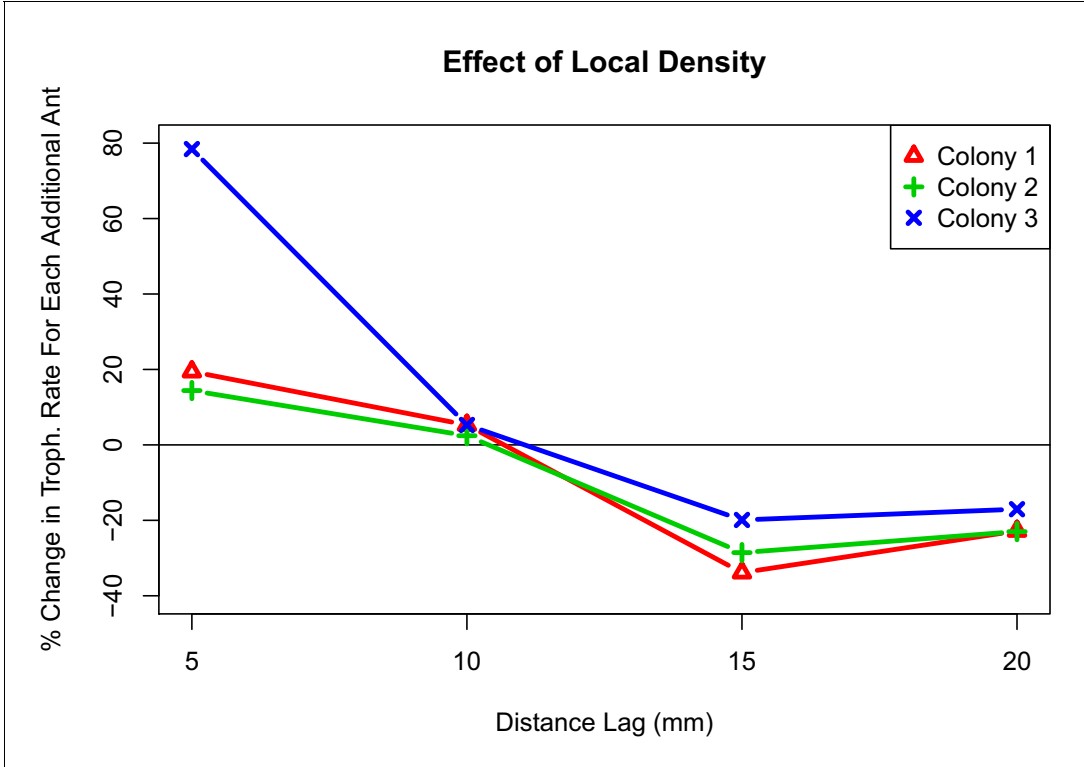

**Figure 3.** Interaction plot depicting the effect of local density on trophallaxis rate over a range of radii. The values shown are the percent change in average rate of trophallaxis events when one additional ant is present at different distance lags. For all colonies, additional ants within 10 mm correlate with increased rate of trophallaxis events, while additional ants within 10 mm–20mm correlate with a decreased rate of trophallaxis events.

DOI: https://doi.org/10.7554/eLife.38473.019

but having additional ants nearby (15 mm or 20 mm) decreased the rate of trophallaxis initiations. Overall, this suggests that ants are more likely to initiate trophallaxis when there are small clusters of ants separated by some additional space. This is a surprising result, because the expectation would be that there are negative effects of density at all radii as the result of crowding, that is, if there are fewer ants around, then the ants will find each other and initiate a trophallaxis event more frequently.

In summary, even though ants had lower observed local density as a result of nest expansion, they did not decrease the rate at which they had trophallaxis interactions. Our data consequently provide empirical support for the hypothesis that ants make a change in their interaction behavior that is independent of local, realized density. By contrast, the results from *Gordon et al. (1993)* suggested that contact via antennation is actively regulated through changes in movement and distribution (regulation of local density). We were able to demonstrate that the regulation of interaction rates cannot be entirely explained by a regulation of local density. In other words, we showed that ants make changes to their interaction behavior that are independent of local density. This difference to the findings of *Gordon et al. (1993)* could be due to the fact that they measured a different form of interaction, that is, antennation versus trophallaxis. In summary, we found that ants were able maintain the rate of their trophallaxis interactions through changes in behavior that were independent of local density.

To better understand how the ants maintain their well-connected network structure, we also examined the underlying movement patterns. We analyzed the tracking data using stochastic differential equation (SDE) models for animal movement (*Russell et al., 2016*; see 'Materials and methods'). These SDE models capture directional persistence in movement through a continuous-time correlated random walk (*Johnson et al., 2008*), directional bias in movement through a spatially varying potential surface (*Brillinger et al., 2002*; *Preisler et al., 2013*), and changes in overall animal movement rate through a spatially varying motility surface (*Russell et al., 2016*). The

**Table 2.** Results of a pair-based analysis of when ant pairs initiate trophallaxis.

We modeled trophallaxis initiations for each pair as coming from an inhomogeneous Poisson point process, where the rate of trophallaxis initiations depends on colony and local-density effects. We included effects for each colony and experimental condition (high or low), and also considered interactions between colony and the local density at 5 mm (n5), 10 mm (n10), 15 mm (n15), and 20 mm (n20).

|  | Estimate | Standard error | Z-value | p-value |
|---|---|---|---|---|
| Colony 1 | −5.80 | 0.07 | −82.25 | <2e-16 |
| Colony 2 | −6.04 | 0.09 | −69.18 | <2e-16 |
| Colony 3 | −8.59 | 0.29 | −30.04 | <2e-16 |
| Treatment | −0.49 | 0.04 | −11.11 | <2e-16 |
| n5 | 0.18 | 0.04 | 4.08 | 4.60e-05 |
| n10 | 0.05 | 0.02 | 2.08 | 0.04 |
| n15 | −0.41 | 0.03 | −15.01 | <2e-16 |
| n20 | −0.26 | 0.02 | −10.57 | <2e-16 |
| col2:n5 | −0.04 | 0.07 | −0.62 | 0.5335 |
| col3:n5 | 0.40 | 0.07 | 5.98 | 2.30e-09 |
| col2:n10 | −0.03 | 0.04 | −0.69 | 0.49 |
| col2:n15 | 0.08 | 0.04 | 1.77 | 0.08 |
| col3:n15 | 0.19 | 0.03 | 5.84 | 5.29e-09 |
| col2:n20 | −0.002 | 0.04 | −0.06 | 0.96 |
| col3:n20 | 0.07 | 0.03 | 2.14 | 0.03 |

DOI: https://doi.org/10.7554/eLife.38473.020

estimated motility surfaces for all colonies (*Figure 1D*, *Figure 1—figure supplement 3*) show that ants move much faster when they travel through the middle two chambers of the enlarged nest. These chambers were predominantly empty. By contrast, ants move much slower in occupied chambers, irrespective of density. The estimated potential surfaces reveal regions in the nest where ants change direction and/or speed consistently over our observation window, with the average force acting on an animal (*Katz et al., 2011*) proportional to the negative gradient of the potential surface (*Preisler et al., 2013*). Thus, animals move predominantly 'downhill' on the potential surface. The estimated potential surfaces (*Figure 1E*, *Figure 1—figure supplement 4*) reveal that ants accelerate upon entering one of the two middle chambers, then change direction quickly as they go around corners in the nest. The change of speed in these middle chambers is striking, implying the ants perceive this space differently. Mechanistically, this may involve learning, the establishment of chemical trails and/or the low rate of encounters with other ants. In summary, workers primarily use the middle two chambers as a corridor for faster travel between the two ant groups. This helps a colony to retain fast flow of information and food within the colony despite the spatial separation.

In conclusion, our results showcase the kinds of behavioral mechanisms that ant colonies apply to achieve social homeostasis in the face of disturbance. Specifically, ants change their spatial distribution, movement dynamics and interaction behavior (independent of local density) in a way that allows them to maintain critical elements of their spatial organization and social interaction patterns despite drastic changes in their environment. This is crucial, because changes in either of these factors are predicted to have large impacts on the efficiency of division of labor and food distribution in the colony (*O'Donnell and Bulova, 2007*).

## Materials and methods

### Ant collection and maintenance

Three queenright *Camponotus pennsylvanicus* colonies were collected in the State Game Lands northeast of State College, PA, USA in Spring 2015. They were subsequently kept in the laboratory

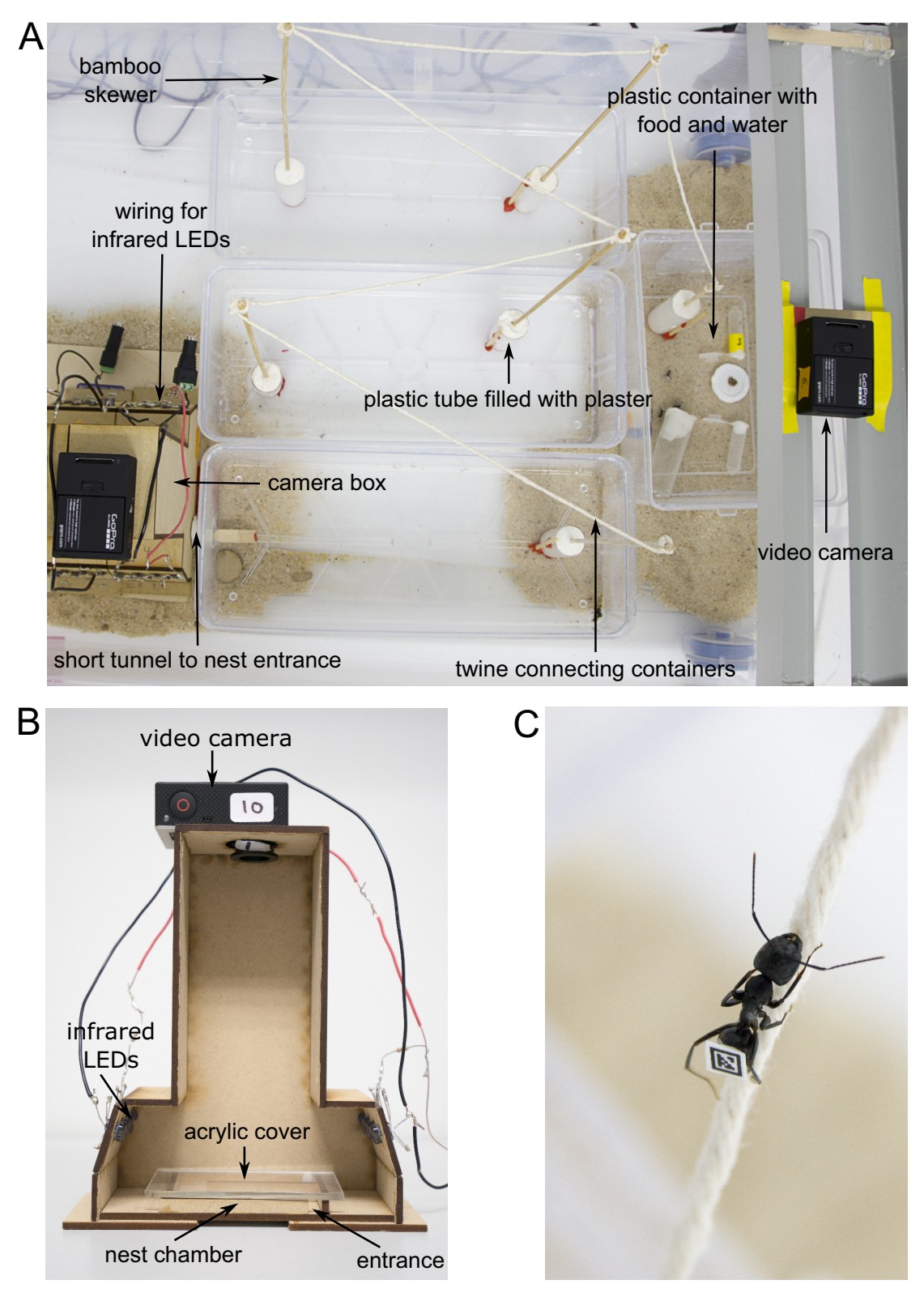

**Figure 4.** Experimental set-up. (**A**) Overview of the foraging arena. (**B**) Camera box with side wall removed to show inner parts. (**C**) Worker walking over twine. Photos by Christoph Kurze.

DOI: https://doi.org/10.7554/eLife.38473.021

under a 14 hr:10 hr light:dark regime at about 24℃, and fed weekly with crickets and 20% sucrose solution.

## Experimental set-up

About 1 week prior to the start of the experiments, we selected a queen, 85 workers and 15 larvae from each colony to create three experimental colonies of comparable colony size. Each worker was tagged with a unique 2 × 2 mm label (*Crall et al., 2015*). The labels were fixed to the gaster of each worker using cyanoacrylate glue (Maxi-Cure, Hobbytown USA), while the ants were immobilized with monofilament line (South Bend, Northbrook, IL) on styrofoam blocks.

After the glue had dried, each colony was allowed to move into an artificial nest made of a wooden U-shaped chamber (65 × 40 × 6 mm) with a transparent acrylic cover and a 6-mm-wide entrance. Once the ants had moved into the nest, the chamber was placed inside a wooden camera box (*Figure 4B*). This allowed us to film ants inside their nest under infrared light from above (distance between camera and nest = 165 mm) using GoPro-cameras (Hero3 and Hero3$^+$ GoPro cameras with modified infrared filters and macro lenses, RageCams, Sparta, MI; Mega IR, 8 mm, 1/2.5"). The nest was illuminated by fourteen 5 mm infrared LEDs (Adafruit Industries, New York, NY; 940 nm wavelength, 20 degree beam width) to maintain the ants in a natural setting of complete darkness, thereby avoiding light disturbance.

Ants were able to leave the nest at any time to enter the foraging arena, which was connected to the nest chamber with a short tunnel made of Sugru self-setting rubber (Sugru Inc, Livonia, MI). The foraging arena was maintained under a 14 hr:10 hr light:dark regime. To better mimic natural foraging distances, the foraging arena consisted of four open plastic containers that are connected in a series: three large elongated containers (30.5 × 13 × 6.5 mm), and one smaller container (19 × 14 × 9.5 mm) with food that was placed furthest away from the nest (*Figure 4A*). To prevent ants from escaping, the walls of the foraging arena were covered with Fluon. To further increase the distance to the food, ants had to climb up bamboo skewers and walk over parcel post twine (Lehigh Group, Macungie, PA) to travel from one container to the next (*Figure 4 A and C*). This resulted in a minimum distance of at least 188 cm from the nest entrance to the food location. A forager is roughly 8 mm meaning the distance covered was 235 times its body length. The food consisted of 20% sucrose solution, water and cricket paste. Cricket paste was prepared by homogenizing three adult crickets with a TissueLyser II (Quiagen, Valencia, CA) using four 1/8" metal bearings (Wheels Manufacturing Inc, Louisville, CO) at a frequency of 24.0 Hz for 66 s. We also placed a GoPro camera with a modified infrared filter above the food to differentiate ants who merely left to nest from actual foragers. The food location was illuminated by a power LED infrared illuminator (CMVISION IRS48 WideAngle; C and M Vision Technologies Inc, Houston, TX) which allowed us to identify foragers at the food when the foraging arena was dark.

It was not feasible to add a temporal control, so the following steps were performed to minimize acclimation effects. First, all colonies were kept in the laboratory for several months to avoid any short-term changes due to the stress of destroying their original home during collection. Second, colonies were given a full week to acclimate after potential disturbance resulting from the set-up of the experiment to ensure that we measured natural behavior. Third, our nests were built to resemble their natural nest sites as much as possible: we used only infrared light and kept them in wooden chambers. Fourth, the foraging arena was set up in a way that forced the ants to show natural foraging behavior: they had to leave the nest and wander quite some distance (188 cm) to find food. All of this was done to minimize acclimation effects resulting from the laboratory setting.

## Video recording

After a week of habituation, we filmed the ants inside their chamber for four hours. We chose to begin filming about 45 min after the start of the dark cycle, because foraging in *C. pennsylvanicus* is primarily nocturnal (*Fowler and Roberts, 1980*). One day after the completion of the filming, we quadrupled the available nest space by adding three more nest chambers (each contained within a camera box) to our set-up. The three new boxes were connected to the original camera box via a 6 mm-wide entrance that allowed the ants to move freely from the original nest chamber through the new chambers into the foraging arena (see *Figure 4A*). After 1 week of habituation, we filmed the four nest chambers for another four hours.

## Collection of trophallaxis and tracking data

Seventeen undergraduate students were recruited and individually trained to record all trophallaxis events inside the nest by noting the start time, end time and the identities of the participating ants. For this, the students individually followed each ant throughout a video and recorded trophallaxis events that were at least two seconds long. Subsequently, the first author re-watched every single trophallaxis event to ensure that the times and identities were correctly recorded by the students. We also manually collected the spatial location of all ants inside the nest for every second of the four-hour observation period for all colonies and treatments. For this, we converted all videos into screenshots with a temporal resolution of one screenshot per second. Students subsequently recorded the position of each ant individually by clicking on the point where the head and thorax ('neck') met on the focal ant in each screenshot of a video. We used a customized Python code (*Colman, 2018*; copy archived at https://github.com/elifesciences-publications/tracking_tool) to automatically record the x- and y-coordinates that were generated by the students' clicking, speeding up this time-consuming process. When the neck of an ant was not visible in a screenshot, because the ant was underneath other ants for instance, we estimated the neck area based on other visible body parts and/or the neck's last known position. To standardize the location data across treatments and nest chambers, we also acquired the coordinates of all nest chamber corners. We subsequently created animations of the spatial movement and trophallaxis events to spot and correct errors (e.g., *Video 1*).

Last but not least, it is worth noting that in the future manual tracking will probably be replaced by automated tracking systems. Knowing first-hand how much work manual tracking involved for this study, we hope that automation or machine learning will continue to improve, and that in the future, these improvements will allow us to reduce the time for data collection by months, if not years (e.g., the new idTracker.ai [http://idtracker.ai/]).

## Data processing

The video recordings with GoPro cameras were continuous, but automatically divided into multiple videos with either 17 min 35 s length (GoPro Hero3) or 26 min ~4.5 s length (GoPro Hero3 silver). After data collection, we accounted for the 0.5 s difference per video between the two camera types by subtracting one second from every second video recorded with a GoPro Hero3 silver. In addition to this adjustment, we also accounted for time delays (a few seconds between cameras in the low-density treatment) that resulted from sequentially turning on multiple cameras. The resulting 'global time' was then used for all further analysis. We confirmed these corrections visually in the actual videos (for instance when ants moved between chambers) and in the animations.

Individual ant locations were first recorded in units of image pixels from the digital camera recording. Affine transformations were used to align these locations with the ant nest by finding the optimal affine transformation relating the corners of each nest chamber to the known dimensions of each nest chamber. Occasionally, the ant locations ended up outside the nest chamber after such a transformation (<1%). The transformed locations were projected onto the nest chamber polygon, and are recorded as being the location in the nest closest to the projected location. When ants move between nest chambers, they are often not visible on any camera for a few seconds. We used linear interpolation to impute ant locations at these times, again projecting any interpolated points that fall outside of the nest to the closest location within the nest. This results in a full set of second by second locations for all ants within the nest during the observation period.

## Classification algorithms for the spatial groups

We developed a novel method to partition a colony into distinct groups on the basis of the similarity of their spatial movements. The algorithm first creates a network in which the weight of the connection between two ants is the similarity of the set of locations that they occupy, it then partitions the ants into separate groups using a standard community-detection algorithm from the network literature (i.e., the Louvain algorithm, *Blondel et al., 2008*). The following paragraphs describe each step in detail.

To quantify the location of an ant at a given point in time, we measured their distance to the entrance, that is how far they would need to move to leave the nest, taking into account the fact that she cannot walk through walls. This is measured as the distance an ant would need to travel to

leave the nest, assuming that their direction of movement is either parallel or perpendicular to the nest walls. We chose this metric because it gives a more realistic path to the entrance than the shortest possible path. The spatial signature of an ant can be characterized by the distribution of all their recorded locations during the 4-hr period.

To measure similarity between two ants, we measured the difference between their location distributions using the Kolmogorov-Smirnov (KS) statistic, which measures the largest distance between the cumulative distributions of the two spatial signatures. The advantage of this measure is that it is invariant to changes in scale, meaning that if both distributions were stretched so that they are, say, twice as wide, then this would not change the value of the KS statistic. We defined similarity between two ants, i and j, as $S_{i,j} = 1–KS(i,j)$, where $KS(i,j)$ is the KS statistic between the distributions of ants i and j. We did this for every pair of ants, resulting in a weighted network for which the weight of the edge is the similarity of the adjacent nodes.

To group the ants, we then performed community detection on the similarity network using the Louvain algorithm (*Blondel et al., 2008*). The algorithm partitions the network into several groups in a way that maximizes the similarity between nodes belonging to the same group while minimizing the similarity between nodes belonging to different groups. In principal, the smallest number of groups that can be detected is 1, this would occur if the similarity we calculate between every pair of ants is exactly the same (but not 0). The largest number of groups possible is equal to the number of ants, this would occur if there is no overlap at all between the ants' spatial signatures.

To test the sensitivity of the outcome of this procedure, we asked whether the identified groups are robust against perturbations in the data. We additionally created 1000 perturbed networks in which the new similarity of each edge was randomly sampled from a beta distribution whose mean is given by its actual similarity (in cases where $S_{i,j} = 0$, the mean was chosen to be 0.001). The shape parameter of the beta distribution was chosen to be $\beta = 4$, which corresponds to mean change in similarity of 10% over all the edges. We applied the community detection algorithm to each of the perturbed networks and compared the group membership of each ant in the actual network to her group in the perturbed equivalent. We consider the ant to have changed group if fewer than half of the ants she was grouped with in the smaller of the two groups are also present in the larger of the two groups. In all cases, fewer than 5% of the ants changed group membership, implying that the detected groups are robust to relatively large corruptions of the data.

## Consistency of the spatial groups

After performing case-wise deletion to remove individuals that died during the experiment, the procedure for identifying groups of ants was applied to each of the six tracking datasets. In the first colony, three groups were identified in the high-density treatment (sizes 35, 22 and 19 ants) and two groups in the low-density treatment (sizes 32 and 41 ants). In the second colony, three groups were identified in the high-density treatment (sizes 35, 24 and 13 ants) and three groups in the low-density treatment (sizes 32, 7 and 41 ants). In the third colony, three groups were identified in the high-density treatment (sizes 23, 24 and 29 ants) and three groups in the low-density treatment (sizes 38, 30 and 15 ants). We then performed Spearman's rank correlation to test whether the spatial group in the high-density treatment predicts the spatial group in the low-density treatment.

Workers that were outside during an entire observation period (2.6%, 10% and 11.6% for colonies 1, 2 and 3, respectively, during high density; 0% for all three colonies during the low-density treatment) were added as spatial group zero. We performed case-wise deletion to remove individuals that died after the first treatment ($-6.4\%$, 0%, $-3.5\%$ for colonies 1, 2 and 3, respectively). We found that an ant's spatial group before the nest expansion predicted its spatial group afterwards (Spearman's rank correlation: $r = 0.26$, $p = 0.03$, $n_{ants} = 73$ for colony 1; $r = 0.47$, $p<0.0001$, $n_{ants} = 80$ for colony 2; and $r = 0.23$, $p=0.03$, $n_{ants} = 83$ for colony 3). P-values were calculated using a permutation test based on Spearman correlations with 40,000 simulations using the 'jmuOutlier' package in R version 3.4.0 (http://www.r-project.org/).

## Quantitative estimate of local density

We define local density of an ant $i$ at time $t$ as the number of ants that are within a 15 mm distance of ant $i$. The distance here is measured as the straightest possible feasible path from one ant to the other, that is the ants can move around walls but they cannot go through them. We compute this

local density for each ant from each colony at 1 s time intervals. These local density measurements are correlated between ants because the local density for each ant at a given time is a function of the location of all other ants at that time. Similarly, these local density measurements are correlated in time, as ant locations are correlated in time.

We obtain uncorrelated local density measurements by first averaging local density over all ants at each time, resulting in a mean local density measurement for each colony at time $t = 1,2, ... ,14,000$. We then subset each of these time series at 25 min time intervals. After doing so, the resulting time series show no temporal autocorrelation (p-value>0.05 for all colonies and densities), and we can thus reasonably treat the resulting mean local density measurements as independent replicates. We fit a mixed effects model to local mean density with a fixed effect for nest set-up (high- or low-density nest), and random effects for colony.

## Comparing ant trophallaxis rates with local density using different radii

To further assess how different radii of local density affect the propensity for trophallaxis events, we considered an analysis of when ant pairs initiate trophallaxis. For this analysis, we assumed that each pair of ants are independent of all pairs. For a given pair of ants, we modeled the observed trophallaxis initiations as events in an inhomogeneous Poisson process (IHPP). An IHPP is a stochastic process for modeling the times when events (such as trophallaxis events) occur. Each event time is independent of all other event times, and the likelihood of seeing events is controlled by a rate function $\lambda(t)$, which can change over time and depend on covariates. For any given interval of time ($I = (t, t + \Delta)$), the number of observed events in the IHPP is distributed as a Poisson random variable with rate $\int_I \lambda(t)dt$. As we have data collected at each second, we can approximate the likelihood of the IHPP with high accuracy by assuming that $\lambda(t)$ is constant for each one-second interval. The likelihood of the observed set of trophallaxis events for one pair of ants is then:

$$\prod_{t=1}^{T} \lambda(t)^{y_t} \exp\{\lambda(t)\}$$

where $y_t = 1$ if the pair of ants begin trophallaxis in the $t$-th second and $y_t = 0$ otherwise. If we model $\lambda(t)$ using covariates with a log link, we can estimate the effect of different covariates on the rate of trophallaxis using Poisson regression.

As the rate of trophallaxis events should be zero whenever ants are too far apart, we set $\lambda(t)=0$ any time ants were greater than 20 mm away from each other. Although ants must be physically closer than 20 mm to have trophallaxis, we included all times when ants were within 20 mm of each other, implicitly assuming that ants are aware of each other and can quickly close this gap and initiate trophallaxis at will with others within this radius. We modeled $\lambda(t)$ using multiple covariates. For each second of observation, we calculated the number of additional ants within 5 mm, 10 mm, 15 mm, and 20 mm of the centroid of the pair of ants. This provides four measures of local density, each at different distance lags. These local density effects were coded as the additional numbers of ants within each successive distance. The focal pair of ants was not included in this calculation, so, for example, when two ants are within 20 mm of the centroid of the focal pair of ants, with one ant being 7 mm away and the other ant being 12 mm away from the centroid, then the local density covariates for 5 mm, 10 mm, 15 mm, and 20 mm were coded as, respectively, 0, 1, 0, and 1. We also included categorical variables for colony (1, 2, 3) and treatment (high- or low-density) in our analysis. Interactions between colony and treatment in this Poisson regression analysis were not significantly different from zero (p-value >0.05, Z-test), and are thus not reported. As there are no significant interaction effects with treatment (high- and low-density), this analysis reveals overall trends in trophallaxis rates that are conserved across colonies.

## Stochastic differential equation modeling of tracking data

We analyzed the tracking data using stochastic differential equation (SDE) models for animal movement (*Russell et al., 2016*). These SDE models capture directional persistence in movement through a continuous-time correlated random walk (*Johnson et al., 2008*), directional bias in movement through a spatially varying potential surface (*Brillinger et al., 2002*; *Preisler et al., 2013*), and changes in overall animal movement rate through a spatially varying motility surface (*Russell et al., 2016*). For simplicity, we will write out our SDE models in one-dimension, with $x_t$ denoting the

location of an ant at time $t$. The ant's velocity at time $t$ is $v_t = \frac{d}{dx}x_t$ and is approximated using $v_t = (x_t - x_{t-1})/\mathrm{h}$, where $h$ is the temporal step size. Our SDE model for movement considers modeling acceleration, which is proportional to the force acting on an ant. Following *Russell et al. (2016)* and *Hanks et al. (2017)*, we model:

$$dv_t = -\beta(x_t - \mu(x_t))dt + c(x_t)dW_t$$

where $\beta$ captures temporal autocorrelation (directional persistence), $\mu(x_t)$ is a spatially varying mean vector, and $c(x_t)$ is a spatially varying variance of the Brownian motion process $W_t$, and can also be seen as a time dilation function (*Hanks et al., 2017*). Taking a second-order approximation (as in *Hanks et al., 2017*) results in a model for position that depends on the position at the previous two time steps:

$$x_t = x_{t-1}(2 - \beta h) + x_{t-2}(\beta h - 1) + \beta h^2 \mu(x_{t-2}) + N\left(0, h^3 c^2(x_{t-2})\right)$$

Under a potential function approach, we set $\mu(x_t) = c(x_t)\frac{d}{dx}P(x_t)$, where $P(x)$ is a potential surface (*Brillinger et al., 2002*; *Preisler et al., 2013*) that controls directional bias in movement. We also allow *c(x)* to vary as a motility surface that affects absolute speed without affecting directional bias, whereas *P(x)* affects both absolute speed and directional bias. We model both $P(x)$ and $c(x)$ using penalized spline expansions, with 2-d spline basis functions being constant on a fine grid, and penalizing the square of the second derivative of both functions. We estimated movement parameters by iterating through the following steps.

1. Assuming a uniform motility surface, we estimated $\beta$ and $P(x)$ using penalized spline fitting implemented in the GAM package in R.
2. Given these estimates, we then estimated the motility surface $c(x)$ using the residuals from step 1. This optimization was again done using GAM in R.
3. Using this estimate for the motility surface, we then re-estimated estimated $\beta$ and $P(x)$ using penalized spline fitting.

The first two steps of this procedure essentially consist of a restricted maximum likelihood (REML) estimate of the motility surface, and the third step estimates the mean parameters (the autocorrelation parameter and the potential surface) conditioned on the REML estimate of the motility surface.

## Data availability

Raw data sets are available through Dryad (DOI: 10.5061/dryad.sh4m4s6). We have compiled the classification algorithms for the spatial groups in the following GitHub repository: https://github.com/EwanColman/Ant-colonies-maintain-social-homeostasis-in-the-face-of-decreased-density (*Colman and Bansal, 2018*; copy archived at https://github.com/elifesciences-publications/Ant-colonies-maintain-social-homeostasis-in-the-face-of-decreased-density). Code to replicate the stochastic differential equation modeling of the tracking data is available in the following GitHub repository: https://github.com/ehanks/Ants-SDE-Motility-Potential (*Hanks, 2018*; copy archived at https://github.com/elifesciences-publications/Ants-SDE-Motility-Potential).

## Acknowledgments

Above all, we would like to thank our undergraduate students at Penn State University who have spent more than 1,600 hr over the past two years manually recording millions of data points of ant behavior and movement: Alyssa Black, Joann Claude, Chad Coakley, Kevin Cosgrove, Amanda Everman, Brianna Good, Krista Grennan, Amelia Hare, Alyssa Kresge, Alyssa Kustenbauder, Amy Luong, Leslie Rowland, Jesse Schneider, Jonah Ulmer, Dieunina Vallon, Torey Vayer and Casey Zipfel. Without their contribution, this project would not have been possible. We are indebted to Deborah Gordon, Ian Baldwin, James D Crall, Jacob Davidson and one anonymous reviewer for their very helpful and detailed comments during the revision process. We thank the Huck Institutes for support. This project was funded by NSF Grant No. 1414296 as part of the joint NSF-NIH-USDA [NIH-NSF-USDA, USDA-NSF-NIH] Ecology and Evolution of Infectious Diseases program.

## Additional information

### Funding

| Funder | Grant reference number | Author |
| --- | --- | --- |
| National Science Foundation | 1414296 | Ephraim M Hanks<br>Shweta Bansal<br>David P Hughes |

The funders had no role in study design, data collection and interpretation, or the decision to submit the work for publication.

### Author contributions

Andreas P Modlmeier, Conceptualization, Data curation, Formal analysis, Supervision, Validation, Investigation, Visualization, Methodology, Writing—original draft, Project administration, Writing—review and editing; Ewan Colman, Data curation, Software, Formal analysis, Validation, Visualization, Methodology, Writing—review and editing; Ephraim M Hanks, Conceptualization, Resources, Data curation, Software, Formal analysis, Supervision, Funding acquisition, Validation, Visualization, Writing—review and editing; Ryan Bringenberg, Supervision, Validation, Investigation, Methodology, Writing—review and editing; Shweta Bansal, Conceptualization, Resources, Supervision, Funding acquisition, Writing—review and editing; David P Hughes, Conceptualization, Resources, Supervision, Funding acquisition, Methodology, Project administration, Writing—review and editing

### Author ORCIDs

Andreas P Modlmeier (iD) http://orcid.org/0000-0002-3095-488X

### Decision letter and Author response

Decision letter https://doi.org/10.7554/eLife.38473.026
Author response https://doi.org/10.7554/eLife.38473.027

## Additional files

### Supplementary files

• Transparent reporting form
DOI: https://doi.org/10.7554/eLife.38473.022

### Data availability

Raw data files have been uploaded to Dryad. Code is available on GitHub.

The following dataset was generated:

| Author(s) | Year | Dataset title | Dataset URL | Database and Identifier |
| --- | --- | --- | --- | --- |
| Modlmeier AP, Colman E | 2018 | Data from: Ant colonies maintain social homeostasis in the face of decreased density | https://dx.doi.org/10.5061/dryad.sh4m4s6 | Dryad, 10.5061/dryad.sh4m4s6 |

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
