## [Decision Letter]

Thank you for submitting your article "Ant colonies maintain social homeostasis in the face of decreased density" for consideration by *eLife*. Your article has been reviewed by 3 peer reviewers, and the evaluation has been overseen by a Reviewing Editor and Ian Baldwin as the Senior Editor. The following individuals involved in review of your submission have agreed to reveal their identity: James D Crall (Reviewer #1); Jacob Davidson (Reviewer #3).

The reviewers have discussed the reviews with one another and the Reviewing Editor has drafted this decision to help you prepare a revised submission.

Summary:

This reports on how interaction networks in ants depend on spatial patterns of movement and local density.

Essential revisions:

The manuscript reports on an impressive amount of data and asks interesting questions. The reviewers raise many issues with statistical analysis and interpretation of the results. A revised version should respond to all of these questions. Without further clarification it is not possible to evaluate the novelty and broad interest of the results.

Reviewer #1:

This paper presents a rich and interesting dataset on the location and movements of uniquely identified ants (*Camponotus pennsylvanicus*) within chambers of varying sizes to investigate the regulation of social interactions as a function of density. The subject is an interesting and timely one, and I think the results will be of broad interest to the social insect and collective behavior communities. Making such a large, manually annotated dataset available on ant movements and locations has the potential to also be an important resource.

I do, however, think there are important points that need to be addressed, that would help strengthen the manuscript, and clarify its interpretation. Perhaps most importantly, I think the authors need to do some work to clarify the hypothesis they are testing, and how their results support/refute this hypothesis. Specifically, the authors find that interaction rate doesn't decrease when density (defined as chamber size) increases. This could result from either (a) a functional change in local density (i.e., interaction rates are a product of local density, but local density is actively regulated by the ants), or (b) a change in interaction behavior that is independent of local, realized density. I don't think the current Results and discussion section clearly distinguishes between these possibilities, but the data collected here offer an opportunity for a more explicit test.

Specifically, I would suggest that the authors use their rich dataset to generate some quantitative estimates of local density, as opposed to overall density within the chambers. This could provide a more explicit test of whether ants are maintaining a consistent local density after the increase in chamber size. There are plenty of approaches for this, including spatial/temporal binning, etc. Gordon et al., 1993, which the authors cite, has some good examples of such an approach (including local v overall interaction rate).

It's also a bit unclear to me what the potential surface modeling (vs. just the intermediate step of motility surface modeling), adds to the paper's conclusions. It seems like the motility surfaces (e.g., in Figure 1—figure supplement 2) provide sufficient support for most of the conclusions in the Results and discussion section (e.g. that ants move fast through the middle chambers, which are lower density and used primarily for transit). In addition, the estimated potential surfaces from this modeling approach appear to show significant variation, both locally within chambers, and between colonies (Figure 1—figure supplement 3), suggesting this approach might be prone to errors and/or sensitive to noise. If there are key conclusions that can only be supported by the potential surface modeling approach, I think these need to be clarified.

Finally, most of the framing of the paper is around a lack of expected change (i.e. we'd expect decreased density to reduce social interaction rate, but we don't find that). The authors in fact found evidence that there was a significant shift after chamber rearrangement, just that this was an increase in interaction rate with decreased density. I think this result needs a bit more interpretation, as well as addressing the sources of variation in interaction rate between colonies driving this pattern.

Reviewer #2:

This study impresses primarily in its scope, and the labour required to generate its conclusions. I think that the results may be of general interest to ant and social insect researchers, but do not consider myself an interaction network specialist.

Specific comments:

Results and discussion section: while the labour involved in this is impressive, it would be good to highlight the use of machine vision and machine learning approaches to this kind of study, at least to inform others lest they believe the only way of generating such datasets is manually! Markerless techniques exist, e.g. idtracker.ai (Polavieja Lab), and with markers such as the QR codes used in the present study, e.g. Mersch et al., (cited); it would be good to discuss why automated approaches were not used or, if they were tried, why they failed.

Results and discussion section: for a possible comparator dataset for this hypothesis, on involvement in an emigration task (*T. albipennis*) see Richardson et al., (2018).

Results and discussion section: is it possible that trophallaxis interaction rates not increasing with density is explainable by identified cliques of ants not finding each other in the crowd?

Results and discussion section: please provide a reference for claimed dyadic nature of trophallaxis.

Subsection “Classification algorithms for the spatial groups”: I found the purpose of the 'bootleg' networks, and their nature, quite opaque – please provide more explanation both of the what and the why.

Subsection “Data availability”: is there a reason why some code is available from the author on request while the remainder is on a repository? It would seem best practice to put all the code there.

Reviewer #3:

In this work the authors analyze the movement and trophallaxis interactions of ants after manipulating the density by introducing extra chambers into the nest. The authors find that the ants adjust their movement to maintain a similar local density and trophallaxis rates following the nest restructuring. Before the change in nest structure, the ants in the nest separated into two groups, and after the change the ants maintained the same group membership structure. I think the results are interesting and should be published. I have some comments of additional points to improve the discussion in the manuscript, and several technical points that need to be addressed prior to publication.

Technical points:

- Subsection “Classification algorithms for the spatial groups”, bootleg. The technical terms in this section are incorrect. I think the authors meant to refer to the "bootstrap" method, instead of the "bootleg" method. However, the procedure they describe is not bootstrapping – it is a method to ask how sensitive the results of the community detection algorithm are to added noise/uncertainty. The procedure they describe is a reasonable way to do this – but it needs to be referred to appropriately.

- Subsection “Stochastic differential equation modeling of tracking data”, SDE coefficients. The authors say "..allow c(x) to vary as a motility function that controls absolute speed." From the equation, this is incorrect – c(x) controls the magnitude of random changes in speed, while mu(x) controls the (spatially-varying) average speed.

Discussion and clarification

- Density and interactions, Results and discussion section. The authors downplay the relationship between spatial movement/density and interactions. Ants cannot interact unless they are close to each other, so this is a clear constraint on possible interactions. Thus, "interaction assortivity" based on spatial locations should be expected, with anything else being surprising. The authors should mention this when they introduce the assortivity measures. Davidson and Gordon, 2017 deals with the distinction between local density and interaction – this could be an interesting comparison for future work with this dataset.

- Mechanisms of maintaining interactions rates with decreased density, Results and discussion section. The authors suggest two possible mechanisms – physiological limitations, and change in spatial structure. The authors confuse density and interaction and implicitly refer to the them as the same thing. There is no evidence otherwise, so I think this is correct. However, it should be made explicit, so that the assumptions are transparent. E.g. "..ants indeed formed spatially separated clusters to maintain an approximately constant local density, such that trophallaxis rates were approximately the same."

- Trophallaxis versus antennal contact. In Gordon et al., 1993, antennal contact is considered when the density is changed. In this paper, the authors analyze trophallaxis events. Did the tracking distinguish events where only antennal contact and not trophallaxis (transfer of liquids) occurred? This might be a point specific to this species of ants. It would be good to mention/clarify this distinction so that previous work can be compared. Also, would the same trends be seen if the analysis used just antennal contact, instead of trophallaxis?

[Editors' note: further revisions were requested prior to acceptance, as described below.]

Thank you for resubmitting your work entitled "Ant colonies maintain social homeostasis in the face of decreased density" for further consideration at *eLife*. Your revised article has been favorably evaluated by Ian Baldwin (Senior Editor), a Reviewing Editor, and three reviewers.

The manuscript has been improved but there are some remaining issues that need to be addressed before acceptance. Both reviewers 1 and 3 suggest further analyses to test whether the rate of trophallaxis depends on local density. They suggest different approaches. A revised version should include some way of addressing these concerns.

Reviewer #1:

Overall, I think the authors have made substantial improvements to the manuscript; in addition to small changes throughout, the authors have done a good job of highlighting the key questions of the manuscript, and how they're answered by these experiments.

However, I think this clarity also highlights some concerns about the specific, central claims of the paper that need to be addressed. In particular, I've got concerns about the data supporting both of the major claims in the introduction. Specifically, the authors now claim that local density (in addition to global density) is reduced in the "low density" treatment. I thank the authors for including the new data, which is very helpful for interpreting results. The authors do a good job of addressing potential temporal autocorrelation in these data, but I don't think a lack of temporal auto-correlation is sufficient evidence for observations from a single colony to be considered independent, since temporal auto-correlation isn't the only source of bias/correlation in these data. As in the analyses of interaction rate (see comments below), I also think it's important to include treatment*colony interactions in statistical models, given that there are clear systematic differences between colonies for other metrics.

Given the rich dataset the authors have collected, I think there may be an alternative analysis that more explicitly addresses what I interpret as the authors' key claim; that ants behaviorally regulate the rate of trophallaxis independent of changes in local density. Specifically, I think the authors could (a) use their data to estimate the probability of a trophallaxis event as a function of pairwise distance between ants (possibly as a logistic regression?), and which should essentially by definition have a strong relationship, and then (b) test whether this relationship differs between density treatments. This would be an explicit test of the hypothesis that trophallaxis behavior varies independent of spatial proximity/local density, and I believe would provide a more convincing test of the authors' key hypotheses.

Reviewer #2:

I think the authors have done a reasonably thorough job of revising the manuscript in light of the three reviewers' comments. However, I feel tracking is very likely to work well for with a state-of-the-art technique for these group sizes, especially given the insects are marked; e.g. idTracker.ai should be cited so for possible validation of these results in the future, and considered by the authors for future work – that group is going as far as automating analysis of interaction networks (http://idtracker.ai/). From my perspective, apart from this issue, I am happy to endorse publication.

Reviewer #3:

Although the authors have addressed many of the issues brought up with the first comments, there are still several remaining points that I feel need to be addressed before publication.

- The authors added the calculation of local density, which I think is a very interesting comparison. However, I feel that the paper currently does not include enough evidence to claim that "ants maintain trophollaxis rates by changing behavior, not by maintaining local density". I would like to see (1) the distribution of local density for individual ants, perhaps according to spatial group, not just the average over the whole group, (2) the correlation between an individual's local density and its trophollaxis rate (e.g. plotting local density versus trophollaxis rate for individual ants), and (3) the results when difference distance cutoffs (both higher and lower than 15mm) are used in the calculation. Regarding (1), because it is mentioned that most ants stay either on one side or the other side, with only a few going back and forth, I might expect that the ants going back and forth are the "density outliers" which bring down the average in the low density treatment case, whereas ants that stay in either area actually have about the same local density in the high vs low density cases. Calculating group local density using the median, and comparing it the current results that use a mean, is another way to see if this may be true. Whether ants change their behavior can be further investigated by (2), because if ants that experience lower local densities change their behavior to maintain trophollaxis rates, there should be no correlation between an ant's average local density and its average trophollaxis rate. Then, to get an idea if the overall interpretation depends on the function used to define local density, it will strengthen the results if the local density calculation is done with different radii. Or if this is not done, the authors should provide justification for the why 15mm is used, and an argument for why the results would not be expected to change if a different distance cutoff, or function for density measurement, is used.

- About the potential surfaces, e.g. Figure 1E. I agree with reviewer 1's comments about the potential surfaces, and that the main conclusions about movement can be obtained more clearly from the motility surface (e.g. Figure 1D). I find the potential surface confusing to look at, because as the authors mention, what it is actually showing is the difference between ants going one direction versus the other direction. I think the trend of moving faster in the middle area is shown well by the motility, and that the potential surface should either be omitted, or calculated separately depending on the departure location (i.e. to calculate two surfaces, one for ants starting from the left side, and one for ants starting from the right side). Without the separation, it seems like because direction and acceleration/deceleration are expected to have opposite trends if going left-right versus right-left, it is not clear to me for example if a moderate value of the potential surface for acceleration/deceleration is representative of actual motion, or is due to averaging. Or if I am interpreting this wrong, please let me know.

- Regarding the spatial groups. There are two or three spatial groups identified by the community detection algorithm, and Figure 2—figure supplement 1 shows nicely how these correspond to the location of trophollaxis locations. However, since the spatial groups are defined by the history of the ants motion, not by the trophollaxis events, it would be informative to show the spatial movement signature associated with each group. For example, something like Figure 4A of Mersch et al., 2013. Without this, I am tempted to use the Figure 2—figure supplement 3 to understand how the different groups have different spatial signatures, but this would actually be incorrect.

[Editors' note: further revisions were requested prior to acceptance, as described below.]

Thank you for resubmitting your work entitled "Ant colonies maintain social homeostasis in the face of decreased density" for further consideration at *eLife*. Your revised article has been favorably evaluated by Ian Baldwin (Senior Editor), a Reviewing Editor, and three reviewers.

The manuscript has been improved but there are some remaining issues that need to be addressed before acceptance, as outlined below:

Reviewer #1:

Overall, I think the additional data and revisions provide and interesting new insight, which will be helpful for interpretation of the paper's key results. However, I think there are a few key points that still need to be addressed, outlined below.

Abstract: The language here suggests that the main finding is a lack of change in interaction rate that is resilient to changes in density. But there's quite strong support for changes in interaction rate, even if in an unexpected direction (either an overall decrease, or divergent effects across colonies, depending on the statistical interpretation, discussed below).

Results and discussion section: I think these new data are really interesting. Am I right that this new analysis, by showing a significant main effect of treatment (Table 2), provides direct support for the idea that the relationship between local density and interaction rate differs between density treatments (suggesting a change in behavior independent of density)? If that interpretation is correct, I think this should be highlighted more clearly in this section, since it provides direct support for a central claim.

Figure 3: I think this figure caption should be expanded to improve interpretability. Are these values derived from the same model estimates in Table 2? If so, specify this in the figure caption. Also, either include units or an expanded discussion of how to interpret the axes.

Table 1: In general, my understanding is that it's not really kosher to interpret overall main effects (e.g., "treatment" or "colony" separately) in the presence of a significant interaction (treatment*colony). In the Results and discussion section (and repeated in the Abstract), the authors suggest that these data support a significant main effect (reduction of interaction rate in high density). Beyond the statistical details, the data Figure 2C clearly suggest substantial differences between colonies in the effect of densities. In light of this, it might be best to clarify why the claim that there is a coherent overall reduction in trophallaxis is justified given these divergent trends across colonies, or remove that claim from the manuscript.

Reviewer #2:

I remain satisfied with the manuscript, but note that the other reviewers have concerns.

Reviewer #3:

I am happy to see that the authors have included an analysis of how trophollaxis rates depend on local density. I think the newly included analysis is a very interesting and important addition to the paper, and is needed to support the claim about regulating trophollaxis rates dependent on density. However, from the methods, I could not follow exactly how this analysis was done. Before publishing, more detail needs to be included in the newly added subsection "Comparing ant trophallaxis rates with local density using different radii", so that the methods can be reproduced.

Here are some questions and my description of how I understood the methods:

Where does the 'inhomogeneous Poisson point process' come into the calculation? I agree that this is a good starting model of trophollaxis events, but I couldn't follow how the calculations were done. The authors say pairs of ants were considered, at times when a neighboring ant (call it N, for neighbor), came into 20 mm of a focal ant (call if F). Then they measured local density around different radii of F, mentioning that they have second-to-second calculations for the different radii. The only way I could think of how a calculation with this description to proceed is to consider all trophollaxis events between F and N, measure the time (call it T_init) from when N reached 20mm of F to when the trophollaxis event occurred, and then to consider the "initiation rate" as 1/T_init. Then, the average local density at the different radii could be used in a regression model to predict initiation rate, given colony, treatment, and local density measurements. Is this what was done? If so, then how does the calculation "For each second in which they were within this proximity to each other," (subsection “Quantitative estimate of local density”), come into play? Wouldn't an average of this local density in the time proceeding the trophollaxis event need to be used? And where does the Poisson process model come into the analysis?

Also, was the neighbor ant N included in the local density counts at the different radii? Since by definition N needs to be close (<5mm) to F in order to perform trophollaxis, it seems more consistent to not include N in the density counts, to avoid any artifacts due to spatial constraints. I think this is what was done, as described (subsection “Quantitative estimate of local density”, "…number of additional ants..")

If the calculation was indeed done as I described above, I would naiively always expect negative effects of density at all radii, due to crowding, i.e. if there are fewer ants around then F and N will find each other and initiate a trophollaxis event quicker. So, then the positive effect of having ants in the "close by" radius of 5mm, which is shown in the paper now, is indeed interesting.

---

## [Author Response]

Essential revisions:The manuscript reports on an impressive amount of data and asks interesting questions. The reviewers raise many issues with statistical analysis and interpretation of the results. A revised version should respond to all of these questions. Without further clarification it is not possible to evaluate the novelty and broad interest of the results.Reviewer #1:This paper presents a rich and interesting dataset on the location and movements of uniquely identified ants (Camponotus pennsylvanicus) within chambers of varying sizes to investigate the regulation of social interactions as a function of density. The subject is an interesting and timely one, and I think the results will be of broad interest to the social insect and collective behavior communities. Making such a large, manually annotated dataset available on ant movements and locations has the potential to also be an important resource.

Thank you. We are glad the reviewer found the work valuable.

I do, however, think there are important points that need to be addressed, that would help strengthen the manuscript, and clarify its interpretation. Perhaps most importantly, I think the authors need to do some work to clarify the hypothesis they are testing, and how their results support/refute this hypothesis. Specifically, the authors find that interaction rate doesn't decrease when density (defined as chamber size) increases. This could result from either (a) a functional change in local density (i.e., interaction rates are a product of local density, but local density is actively regulated by the ants), or (b) a change in interaction behavior that is independent of local, realized density. I don't think the current Results section and discussion section clearly distinguishes between these possibilities, but the data collected here offer an opportunity for a more explicit test.Specifically, I would suggest that the authors use their rich dataset to generate some quantitative estimates of local density, as opposed to overall density within the chambers. This could provide a more explicit test of whether ants are maintaining a consistent local density after the increase in chamber size. There are plenty of approaches for this, including spatial/temporal binning, etc. Gordon et al., 1993, which the authors cite, has some good examples of such an approach (including local v overall interaction rate).

We thank the reviewer for this concise and very helpful suggestion. We completely agree and have subsequently generated a quantitative estimate for local density (please see below for methods) that allows us to differentiate between the two possibilities of how ants maintain interaction rates.

We added the following information to the Results and discussion section:

“…ants may be able to actively regulate their interaction rates to keep them at an optimal level for the colony by (a) changing their movement and distribution in the nest (Adler & Gordon, 1992; Gordon et al., 1993; Davidson and Gordon, 2017), thus maintaining local density, or by (b) making a change in their interaction behavior that is independent of local (realized) density. To examine the relative support for these two scenarios in our experimental data, we examined the local (realized) density for ants in each colony (please see Materials and methods for details). We estimated local density to be, on average, 9.9 ants within 15mm in high-density nests, and 7.9 ants within 15mm in low-density nests. This difference was significant (Wald Chi-square test, p < 0.00001), indicating that local density was indeed lower for ants in low-density nests than for ants in high-density nests. Thus, even though ants had lower observed local density, they did not decrease the rate at which they had trophallaxis interactions. Our data consequently provides empirical support for the hypothesis that ants make a change in their interaction behavior that is independent of local, realized density. In other words, ants were able maintain the rate of their trophallaxis interactions through changes in behavior that were independent of local density.”

The following information was added to the Materials and methods section:

“We define local density of an ant i at time t as the number of ants that are within a 15mm distance of ant i. The distance here is measured as the straightest possible feasible path from one ant to the other, i.e. they can move around walls but they cannot go through them. We compute this local density for each ant from each colony at 1 second time intervals. These local density measurements are correlated between ants as the local density for each ant at a given time is a function of the location of all other ants at that time. Similarly, these local density measurements are correlated in time, as ant locations are correlated in time. We obtain uncorrelated local density measurements by first averaging local density over all ants at each time, resulting in a mean local density measurement for each colony at time t=1,2,…,14,000. We then subset each of these time series at 25 minute time intervals. After doing so, the resulting time series show no temporal autocorrelation (p-value> 0.05 for all colonies and densities), and we can thus reasonably treat the resulting mean local density measurements as independent replicates. We fit a mixed effects model to local mean density with a fixed effect for nest setup (high or low-density nest), and random effects for colony.”

It's also a bit unclear to me what the potential surface modeling (vs. just the intermediate step of motility surface modeling), adds to the paper's conclusions. It seems like the motility surfaces (e.g., in Figure 1—figure supplement 2) provide sufficient support for most of the conclusions in the Results and discussion section (e.g. that ants move fast through the middle chambers, which are lower density and used primarily for transit). In addition, the estimated potential surfaces from this modeling approach appear to show significant variation, both locally within chambers, and between colonies (Figure 1—figure supplement 3), suggesting this approach might be prone to errors and/or sensitive to noise. If there are key conclusions that can only be supported by the potential surface modeling approach, I think these need to be clarified.

You are correct that most of our scientific conclusions are focused on the estimated motility surfaces, which explain in part how ants maintain connectivity while spatially dispersed. However, modeling directional bias through the potential surfaces is critical to obtaining valid estimates of motility surfaces. We know ants have clear directional bias in their movements – for example they change direction quickly when rounding corners in the middle chambers of the nest. Failing to account for such behavior could potentially bias our estimates of the motility surfaces, just like failing to include an important covariate could bias the results of any analysis (i.e., a regression with an important covariate missing).

You are also correct that there are visible differences in the estimated potential surfaces between colonies. We have added additional text explaining the visible differences to the legend for figure 1—figure supplement 4. Some important general patterns are maintained in all colonies, with potential surfaces pushing ants to turn quickly around corners. The major differences between colonies are the differences in the estimated levels of the potential surface in the end chambers. This is caused by observing different numbers of ants moving from right to left vs. from left to right. If there are more ants observed moving from left to right than from right to left in the 4-hour observation window (as in Colony 2), then the estimated potential surface reflects this by having a much higher potential surface on the left than on the right. Beyond this difference, the potential surfaces for the three colonies show nearly the same pattern.

Finally, most of the framing of the paper is around a lack of expected change (i.e. we'd expect decreased density to reduce social interaction rate, but we don't find that). The authors in fact found evidence that there was a significant shift after chamber rearrangement, just that this was an increase in interaction rate with decreased density. I think this result needs a bit more interpretation, as well as addressing the sources of variation in interaction rate between colonies driving this pattern.

Thank you. We have now addressed this in the manuscript.

Reviewer #2:This study impresses primarily in its scope, and the labour required to generate its conclusions. I think that the results may be of general interest to ant and social insect researchers, but do not consider myself an interaction network specialist.

We are happy to receive this positive assessment of our work.

Specific comments:Results and discussion section: while the labour involved in this is impressive, it would be good to highlight the use of machine vision and machine learning approaches to this kind of study, at least to inform others lest they believe the only way of generating such datasets is manually! Markerless techniques exist, e.g. idtracker.ai (Polavieja Lab), and with markers such as the QR codes used in the present study, e.g. Mersch et al., (cited); it would be good to discuss why automated approaches were not used or, if they were tried, why they failed.

We indeed tried automated approaches, but could not get results that would yield acceptable accuracies. Initial results of the tracking resulted in tags only being visible about half of the time. This was presumably due to the fact that tags were hidden when ants moved sideways along the walls, upside down and below other ants. Hence, tags were only visible about half of the time. Due to time constraints, the need for high quality data and the fact that no machine learning expert was part of our research group, we very quickly decided to start tracking manually to quickly get high quality data for the development of spatial models.

Results and discussion section: for a possible comparator dataset for this hypothesis, on involvement in an emigration task (T. albipennis) see Richardson et al., (2018).

We thank the reviewer for this suggestion. We have now added it to the corresponding sentence (Results and discussion section).

Results and discussion section: is it possible that trophallaxis interaction rates not increasing with density is explainable by identified cliques of ants not finding each other in the crowd?

This could indeed be another possible explanation. We added it to the explanations.

Results and discussion section: please provide a reference for claimed dyadic nature of trophallaxis.

The reference for this is personal observation during our experiment. We added this to the manuscript (Results and discussion section). We rarely observed interactions between more than two ants. In some instances, a third ant would come in and eat parts of the food that was shared between the original two ants, but trophallaxis events always started between two ants with a third ant joining in rare instances. Hence, we stated that in rare instances an individual ant would share food with two other ants, i.e., triadic interaction.

Subsection “Classification algorithms for the spatial groups”: I found the purpose of the 'bootleg' networks, and their nature, quite opaque – please provide more explanation both of the what and the why.

The term we meant to use was "Bootstrap”. Apologies for the confusion. Based on the reviewer’s comments we have now changed this to "perturbed" throughout the text. The purpose of this analysis is to test the sensitivity of our results to uncertainty in our data. For example, one concern we had was that an ant could be identified as belonging to, say, group 1, however had we collected an extra hour of data we might find that they belong to group 2.

We have added the following explanation to the material and methods: "To test the sensitivity of the outcome of this procedure, we asked whether the identified groups are robust against perturbations in the data." (Subsection “Classification algorithms for the spatial groups”).

Subsection “Data availability”: is there a reason why some code is available from the author on request while the remainder is on a repository? It would seem best practice to put all the code there.

The reviewer is correct. We have now placed all code to replicate all analyses in the paper on GitHub, and so indicate in the paper.

Reviewer #3:In this work the authors analyze the movement and trophallaxis interactions of ants after manipulating the density by introducing extra chambers into the nest. The authors find that the ants adjust their movement to maintain a similar local density and trophallaxis rates following the nest restructuring. Before the change in nest structure, the ants in the nest separated into two groups, and after the change the ants maintained the same group membership structure. I think the results are interesting and should be published. I have some comments of additional points to improve the discussion in the manuscript, and several technical points that need to be addressed prior to publication.

Thank you for this positive assessment of our work.

Technical points:- Subsection “Classification algorithms for the spatial groups”, bootleg. The technical terms in this section are incorrect. I think the authors meant to refer to the "bootstrap" method, instead of the "bootleg" method. However, the procedure they describe is not bootstrapping – it is a method to ask how sensitive the results of the community detection algorithm are to added noise/uncertainty. The procedure they describe is a reasonable way to do this – but it needs to be referred to appropriately.

Yes, the reviewer is completely correct. We have now removed all bootleg mentions and replaced them with a definition and the term “perturbed”.

- Subsection “Stochastic differential equation modeling of tracking data”, SDE coefficients. The authors say "..allow c(x) to vary as a motility function that controls absolute speed." From the equation, this is incorrect – c(x) controls the magnitude of random changes in speed, while mu(x) controls the (spatially-varying) average speed.

We were imprecise here. We have added additional explanation in the text to show that P(x) affects absolute speed and directional bias, while c(x) only affects absolute speed. See Hanks et al., (2017) and Russell et al., (2018) for details.

Discussion and clarification- Density and interactions, Results and discussion section. The authors downplay the relationship between spatial movement/density and interactions. Ants cannot interact unless they are close to each other, so this is a clear constraint on possible interactions. Thus, "interaction assortivity" based on spatial locations should be expected, with anything else being surprising. The authors should mention this when they introduce the assortivity measures. Davidson and Gordon, 2017 deals with the distinction between local density and interaction – this could be an interesting comparison for future work with this dataset.

We agree with the reviewer and have added an additional sentence to stress that interactions require spatial proximity.

“Due to the spatial fidelity of individual ants, we expected positive assortativity, because ants can only interact with ants that are close to them.” (Results and discussion section).

- Mechanisms of maintaining interactions rates with decreased density, Results and discussion section. The authors suggest two possible mechanisms – physiological limitations, and change in spatial structure. The authors confuse density and interaction and implicitly refer to the them as the same thing. There is no evidence otherwise, so I think this is correct. However, it should be made explicit, so that the assumptions are transparent. E.g. "..ants indeed formed spatially separated clusters to maintain an approximately constant local density, such that trophallaxis rates were approximately the same."

This paragraph was changed (and the corresponding sentence deleted) due to the addition of the new local density measure allowing us to provide support for the hypothesis that ants regulate their interaction behavior independent of local (realized) density.

- Trophallaxis versus antennal contact. In Gordon et al., 1993, antennal contact is considered when the density is changed. In this paper, the authors analyze trophallaxis events. Did the tracking distinguish events where only antennal contact and not trophallaxis (transfer of liquids) occurred? This might be a point specific to this species of ants. It would be good to mention/clarify this distinction so that previous work can be compared. Also, would the same trends be seen if the analysis used just antennal contact, instead of trophallaxis?

Our trophallaxis observations only marked actual transfer of food, we did not collect data of simple contacts via antennae. We would expect that contacts that do not include food sharing events are much more common than interactions that include food sharing, so there would certainly be differences in the frequency. Other changes might also be possible, but this would require actual tests and would thus be an interesting avenue for future research.

[Editors' note: further revisions were requested prior to acceptance, as described below.]

The manuscript has been improved but there are some remaining issues that need to be addressed before acceptance. Both reviewers 1 and 3 suggest further analyses to test whether the rate of trophallaxis depends on local density. They suggest different approaches. A revised version should include some way of addressing these concerns.Reviewer #1:Overall, I think the authors have made substantial improvements to the manuscript; in addition to small changes throughout, the authors have done a good job of highlighting the key questions of the manuscript, and how they're answered by these experiments.However, I think this clarity also highlights some concerns about the specific, central claims of the paper that need to be addressed. In particular, I've got concerns about the data supporting both of the major claims in the introduction. Specifically, the authors now claim that local density (in addition to global density) is reduced in the "low density" treatment. I thank the authors for including the new data, which is very helpful for interpreting results. The authors do a good job of addressing potential temporal autocorrelation in these data, but I don't think a lack of temporal auto-correlation is sufficient evidence for observations from a single colony to be considered independent, since temporal auto-correlation isn't the only source of bias/correlation in these data. As in the analyses of interaction rate (see comments below), I also think it's important to include treatment*colony interactions in statistical models, given that there are clear systematic differences between colonies for other metrics.

We completely agree with the reviewer. Hence, we have now included treatment by colony effects in the new analyses done. In the movement analyses, as each was conducted on each colony independent of all others, this allows for individual colony level effects and interactions with all model parameters, as each model parameter is estimated for each colony, independent of all other colonies.

Given the rich dataset the authors have collected, I think there may be an alternative analysis that more explicitly addresses what I interpret as the authors' key claim; that ants behaviorally regulate the rate of trophallaxis independent of changes in local density. Specifically, I think the authors could (a) use their data to estimate the probability of a trophallaxis event as a function of pairwise distance between ants (possibly as a logistic regression?), and which should essentially by definition have a strong relationship, and then (b) test whether this relationship differs between density treatments. This would be an explicit test of the hypothesis that trophallaxis behavior varies independent of spatial proximity/local density, and I believe would provide a more convincing test of the authors' key hypotheses.

We thank the reviewer for this suggestion, and have conducted a completely new analysis of the individual interaction data based off of your comment. Please see below:

“To get a clearer picture of the effect of local density, we also performed additional tests that included different radii at a range of lags (5mm, 10mm, 15mm, and 20mm), treatment and colony identity (please see Materials and methods for details). We found that colony 3 had a lower baseline rate of trophallaxis initiation than the other two colonies (z-test, p-value<0.001; Table 2), which were not significantly different from each other. Overall, we found lower rates of trophallaxis in the high-density treatment, than in low density. In addition to this treatment-level effect, we found that local (realized) density had significant effects on trophallaxis initiation rates, and that this effect varied slightly between colonies. While there were significant interaction effects between local density and colony, the qualitative effect of local density is very consistent across the three colonies (see Figure 3): the more ants that are very close (5mm or 10mm) to an ant pair, the higher the rate of trophallaxis initiations between pairs, but having additional ants nearby (15mm or 20mm) decreased the rate of trophallaxis initiations. Overall, this suggests that ants are more likely to initiate trophallaxis when there are small clusters of ants separated by some additional space. This is consistent with our finding that ants exhibited lower trophallaxis rates in the high-density treatment, in which all ants are very tightly packed together.”

Reviewer #2:I think the authors have done a reasonably thorough job of revising the manuscript in light of the three reviewers' comments. However, I feel tracking is very likely to work well for with a state-of-the-art technique for these group sizes, especially given the insects are marked; e.g. idTracker.ai should be cited so for possible validation of these results in the future, and considered by the authors for future work – that group is going as far as automating analysis of interaction networks (http://idtracker.ai/). From my perspective, apart from this issue, I am happy to endorse publication.

We thank the reviewer for these kind words and the endorsement for publication. We now mention “idTracker.ai” in our Materials and methods section and are looking forward to times when automated tracking will minimize the time needed to achieve high quality data on a larger scale.

Reviewer #3:Although the authors have addressed many of the issues brought up with the first comments, there are still several remaining points that I feel need to be addressed before publication.

We are glad we were able to address many of the issues and thankful for the provided suggestions to clear up the remaining points.

- The authors added the calculation of local density, which I think is a very interesting comparison. However, I feel that the paper currently does not include enough evidence to claim that "ants maintain trophollaxis rates by changing behavior, not by maintaining local density". I would like to see (1) the distribution of local density for individual ants, perhaps according to spatial group, not just the average over the whole group, (2) the correlation between an individual's local density and its trophollaxis rate (e.g. plotting local density versus trophollaxis rate for individual ants), and (3) the results when difference distance cutoffs (both higher and lower than 15mm) are used in the calculation. Regarding (1), because it is mentioned that most ants stay either on one side or the other side, with only a few going back and forth, I might expect that the ants going back and forth are the "density outliers" which bring down the average in the low density treatment case, whereas ants that stay in either area actually have about the same local density in the high vs low density cases. Calculating group local density using the median, and comparing it the current results that use a mean, is another way to see if this may be true. Whether ants change their behavior can be further investigated by (2), because if ants that experience lower local densities change their behavior to maintain trophollaxis rates, there should be no correlation between an ant's average local density and its average trophollaxis rate. Then, to get an idea if the overall interpretation depends on the function used to define local density, it will strengthen the results if the local density calculation is done with different radii. Or if this is not done, the authors should provide justification for the why 15mm is used, and an argument for why the results would not be expected to change if a different distance cutoff, or function for density measurement, is used.

Based on this comment, as well as comments from reviewer 1, we have conducted a completely new analysis in which we compare ant trophallaxis rates with local density, using different radii as suggested. Please see our responses to reviewer 1 and/or the Results and discussion section for the revised text.

- About the potential surfaces, e.g. Figure 1E. I agree with reviewer 1's comments about the potential surfaces, and that the main conclusions about movement can be obtained more clearly from the motility surface (e.g. Figure 1D). I find the potential surface confusing to look at, because as the authors mention, what it is actually showing is the difference between ants going one direction versus the other direction. I think the trend of moving faster in the middle area is shown well by the motility, and that the potential surface should either be omitted, or calculated separately depending on the departure location (i.e. to calculate two surfaces, one for ants starting from the left side, and one for ants starting from the right side). Without the separation, it seems like because direction and acceleration/deceleration are expected to have opposite trends if going left-right versus right-left, it is not clear to me for example if a moderate value of the potential surface for acceleration/deceleration is representative of actual motion, or is due to averaging. Or if I am interpreting this wrong, please let me know.

We agree that the main result that ants move quickly through the middle chambers is captured best by the motility surface. However, if we do not model the potential surface as well, when we simulate ant movement from the fitted model, the ants move in completely unrealistic ways, as they do not make any attempt to change directions and go around the corners. One way to think of this is that “motility” is the covariate of scientific interest, but “potential” is another confounding variable. We couldn't trust our estimate of motility if we didn't also account for the ants' propensity to change directions quickly around corners. So, our analysis captures both processes, and results in estimates of motility that account for the changes in directional movement.

- Regarding the spatial groups. There are two or three spatial groups identified by the community detection algorithm, and Figure 2—figure supplement 1 shows nicely how these correspond to the location of trophollaxis locations. However, since the spatial groups are defined by the history of the ants motion, not by the trophollaxis events, it would be informative to show the spatial movement signature associated with each group. For example, something like Figure 4A of Mersch et al., 2013. Without this, I am tempted to use the Figure 2—figure supplement 3 to understand how the different groups have different spatial signatures, but this would actually be incorrect.

We have now added three new figure supplements that show all ant locations during the observation period for each colony and treatment. The new figures are: Figure 2—figure supplement 4, Figure 2—figure supplement 5 and Figure 2—figure supplement 6.

[Editors' note: further revisions were requested prior to acceptance, as described below.]

The manuscript has been improved but there are some remaining issues that need to be addressed before acceptance, as outlined below:Reviewer #1:Overall, I think the additional data and revisions provide and interesting new insight, which will be helpful for interpretation of the paper's key results. However, I think there are a few key points that still need to be addressed, outlined below.Abstract: The language here suggests that the main finding is a lack of change in interaction rate that is resilient to changes in density. But there's quite strong support for changes in interaction rate, even if in an unexpected direction (either an overall decrease, or divergent effects across colonies, depending on the statistical interpretation, discussed below).

We agree with the reviewer and now have made it clear that there are divergent effects across colonies, contrary to the expectation that there is coherent reduction in interaction rates across all colonies. Please also see our response to Table 1 below.

Abstract:

“Despite a reduction in both overall and local density, we did not find the expected concomitant reduction in interaction rates across all colonies. Instead, we found divergent effects across colonies.”

Results and discussion section: I think these new data are really interesting. Am I right that this new analysis, by showing a significant main effect of treatment (Table 2), provides direct support for the idea that the relationship between local density and interaction rate differs between density treatments (suggesting a change in behavior independent of density)? If that interpretation is correct, I think this should be highlighted more clearly in this section, since it provides direct support for a central claim.

This is correct. We have now added additional text on this analysis to make our methods, and the resulting conclusions, more clear. Moreover, we have incorporated parts of the reviewer’s suggestion into the paragraph (see last sentence below and Results and discussion section in the manuscript).

“To get a clearer picture of the effect of local density, we also performed an analysis of when trophallaxis events occur between pairs of ants. By viewing trophallaxis events as arising from an inhomogeneous Poisson process, we examined how rates of trophallaxis initiations differed across colonies (1,2,3), treatments (high and low density), and also for different levels of “local density”. We defined local density as the number of additional ants (beyond the focal pair) within a range of spatial lags (5mm, 10mm, 15mm, and 20mm). This allows us to examine how much of the variation in trophallaxis rates can be attributed to differences in colony, treatment, and local density. We found no significant interactions (p-value >0.05, Z-test) in this analysis between colony and treatment (high or low density). We found significant interactions between colony and local density (numbers of ants within different spatial lags), but the qualitative patterns in the effects of local density were conserved across all colonies (see Figure 3). Furthermore, we found lower rates of trophallaxis in the high-density treatment than in low density (Table 2). This significant main effect of treatment provides direct support for the idea that the relationship between local density and interaction rate differs between density treatments suggesting a change in behavior independent of density.”

Figure 3: I think this figure caption should be expanded to improve interpretability. Are these values derived from the same model estimates in Table 2? If so, specify this in the figure caption. Also, either include units or an expanded discussion of how to interpret the axes.

Thank you for your suggestion. We have updated the figure and its description.

Table 1: In general, my understanding is that it's not really kosher to interpret overall main effects (e.g., "treatment" or "colony" separately) in the presence of a significant interaction (treatment*colony). In the Results and discussion section (and repeated in the Abstract), the authors suggest that these data support a significant main effect (reduction of interaction rate in high density). Beyond the statistical details, the data Figure 2C clearly suggest substantial differences between colonies in the effect of densities. In light of this, it might be best to clarify why the claim that there is a coherent overall reduction in trophallaxis is justified given these divergent trends across colonies, or remove that claim from the manuscript.

In response to the reviewer’s comment, we have now removed the claims that there is coherent overall reduction in trophallaxis and/or replaced it with “divergent trends across colonies”.

Abstract:

“Despite a reduction in both overall and local density, we did not find the expected concomitant reduction in interaction rates across all colonies. Instead, we found divergent effects across colonies.”.

Results and discussion section:

“The significant interaction between colony and treatment suggests divergent trends among colonies (Table 1).”

Reviewer #2:I remain satisfied with the manuscript, but note that the other reviewers have concerns.

We thank the reviewer. We are glad our changes have been satisfactory and did not raise new concerns.

Reviewer #3:I am happy to see that the authors have included an analysis of how trophollaxis rates depend on local density. I think the newly included analysis is a very interesting and important addition to the paper, and is needed to support the claim about regulating trophollaxis rates dependent on density.

We completely agree. We think that the addition of local density has tremendously improved our manuscript and are thankful for the suggestion to include a range of radii.

However, from the methods, I could not follow exactly how this analysis was done. Before publishing, more detail needs to be included in the newly added subsection "Comparing ant trophallaxis rates with local density using different radii", so that the methods can be reproduced.Here are some questions and my description of how I understood the methods:Where does the 'inhomogeneous Poisson point process' come into the calculation? I agree that this is a good starting model of trophollaxis events, but I couldn't follow how the calculations were done. The authors say pairs of ants were considered, at times when a neighboring ant (call it N, for neighbor), came into 20 mm of a focal ant (call if F). Then they measured local density around different radii of F, mentioning that they have second-to-second calculations for the different radii. The only way I could think of how a calculation with this description to proceed is to consider all trophollaxis events between F and N, measure the time (call it T_init) from when N reached 20mm of F to when the trophollaxis event occurred, and then to consider the "initiation rate" as 1/T_init. Then, the average local density at the different radii could be used in a regression model to predict initiation rate, given colony, treatment, and local density measurements. Is this what was done? If so, then how does the calculation "For each second in which they were within this proximity to each other," (subsection “Quantitative estimate of local density”), come into play? Wouldn't an average of this local density in the time proceeding the trophollaxis event need to be used? And where does the Poisson process model come into the analysis?Also, was the neighbor ant N included in the local density counts at the different radii? Since by definition N needs to be close (<5mm) to F in order to perform trophollaxis, it seems more consistent to not include N in the density counts, to avoid any artifacts due to spatial constraints. I think this is what was done, as described (subsection “Quantitative estimate of local density”, "…number of additional ants..")

We have added much more information both to the main text and the Materials and methods section “Comparing ant trophallaxis rates with local density using different radii” to clarify exactly how we conducted this analysis, our choice of methods, and how to interpret the results. Please see below for the added information and where it can be found in the manuscript.

Results and discussion section:

“To get a clearer picture of the effect of local density, we also performed an analysis of when trophallaxis events occur between pairs of ants. By viewing trophallaxis events as arising from an inhomogeneous Poisson process, we examined how rates of trophallaxis initiations differed across colonies (1,2,3), treatments (high and low density), and also for different levels of “local density”. We defined local density as the number of additional ants (beyond the focal pair) within a range of spatial lags (5mm, 10mm, 15mm, and 20mm). This allows us to examine how much of the variation in trophallaxis rates can be attributed to differences in colony, treatment, and local density. We found no significant interactions (p-value >0.05, Z-test) in this analysis between colony and treatment (high or low density). We found significant interactions between colony and local density (numbers of ants within different spatial lags), but the qualitative patterns in the effects of local density were conserved across all colonies (see Figure 3). Furthermore, we found lower rates of trophallaxis in the high-density treatment than in low density (Table 2).”

Materials and method section:

“To further assess how different radii of local density affect the propensity for trophallaxis events, we considered an analysis of when ant pairs initiate trophallaxis. For this analysis we assumed that each pair of ants are independent of all pairs. For a given pair of ants, we modeled the observed trophallaxis initiations as events in an inhomogeneous Poisson process (IHPP). An IHPP is a stochastic process for modeling the times when events (such as trophallaxis events) occur. Each event time is independent of all other event times, and the likelihood of seeing events is controlled by a rate function λt, which can change over time and depend on covariates. For any given interval of timeI=t,t+Δ, the number of observed events in the IHPP is distributed as a Poisson random variable with rate∫∫Iλ(t)dt. As we have data collected at each second, we can approximate the likelihood of the IHPP with high accuracy by assuming that λt is constant on each 1-second interval. The likelihood of the observed set of trophallaxis events for one pair of ants is then

∏t=1Tλtytexp⁡{λt}

Where yt=1 if the pair of ants begin trophallaxis in the *t-*th second and yt=0 otherwise. If we model λtusing covariates with a log link, we can estimate the effect of different covariates on the rate of trophallaxis using Poisson regression.

As the rate of trophallaxis events should be zero whenever ants are too far apart, we set λt=0 any time ants were greater than 20mm of each other. While physically ants must be closer than 20mm to have trophallaxis, we included all times when ants were within 20mm of each other, implicitly assuming that ants are aware of each other and can quickly close this gap and initiate trophallaxis at will with others within this radius. We modeled λt using multiple covariates. For each second of observation, we calculated the number of additional ants within 5mm, 10mm, 15mm, and 20mm of the centroid of the pair of ants. This provides four measures of local density, each at different distance lags. These local density effects were coded as the additional numbers of ants within each successive distance. The focal pair of ants was not included in this calculation, so, for example, when two ants are within 20mm of the centroid of the focal pair of ants, with one ant being 7mm away and the other ant being 12mm away from the centroid, then the local density covariates for 5, 10, 15, and 20mm were coded as, respectively, 0, 1, 0, and 1. We also included categorical variables for colony (1, 2, 3) and treatment (high or low density) in our analysis. Interactions between colony and treatment in this Poisson regression analysis were found to be not significantly different from zero (p-value >0.05, Z-test), and are thus not reported. As there are no significant interaction effects with treatment (high and low density), this analysis reveals overall trends in trophallaxis rates that are conserved across colonies.”

If the calculation was indeed done as I described above, I would naively always expect negative effects of density at all radii, due to crowding, i.e. if there are fewer ants around then F and N will find each other and initiate a trophollaxis event quicker. So, then the positive effect of having ants in the "close by" radius of 5mm, which is shown in the paper now, is indeed interesting.

We completely agree with this interpretation and thank the reviewer for pointing this out. We now emphasize that our result is surprising given that the expectation would be a negative effect across all radii, due to crowding (Results and discussion section).

“This is a surprising result, because the expectation would be that there are negative effects of density at all radii, due to crowding, i.e., if there are fewer ants around then ants will find each other and initiate a trophallaxis event more frequently.”